# Coalescing beneficial host and deleterious antiparasitic actions as an antischistosomal strategy

John D Chan[1], Timothy A Day[1], Jonathan S Marchant[2]*

[1]Department of Biomedical Sciences, Iowa State University, Ames, United States; [2]Department of Cell Biology, Neurobiology and Anatomy, Medical College of Wisconsin, Milwaukee, United States

**Abstract** Conventional approaches for antiparasitic drug discovery center upon discovering selective agents that adversely impact parasites with minimal host side effects. Here, we show that agents with a broad polypharmacology, often considered 'dirtier' drugs, can have unique efficacy if they combine deleterious effects on the parasite with beneficial actions in the host. This principle is evidenced through a screen for drugs to treat schistosomiasis, a parasitic flatworm disease that impacts over 230 million people. A target-based screen of a *Schistosoma* serotoninergic G protein coupled receptor yielded the potent agonist, ergotamine, which disrupted worm movement. In vivo, ergotamine decreased mortality, parasite load and intestinal egg counts but also uniquely reduced organ pathology through engagement of host GPCRs that repressed hepatic stellate cell activation, inflammatory damage and fibrosis. The unique ability of ergotamine to engage both host and parasite GPCRs evidences a future strategy for anthelmintic drug design that coalesces deleterious antiparasitic activity with beneficial host effects.

DOI: https://doi.org/10.7554/eLife.35755.001

*For correspondence:
JMarchant@mcw.edu

**Competing interests:** The authors declare that no competing interests exist.

## Introduction

Schistosomiasis (infection with parasitic blood flukes) affects over 230 million people, with severe infections causing ~200,000 deaths per year. Chronic infections are associated with morbidities such as anemia, impaired cognitive development and stunted growth (*Colley et al., 2014*; *King and Dangerfield-Cha, 2008*). Infection is acquired upon exposure to freshwater, larval cercariae which penetrate the host skin. Inside host tissue, cercariae transform into schistosomula that access and traverse the circulatory system, pass through the lungs and liver before finally residing as mature adults within either the mesenteric vessels (*Schistosoma mansoni* and *Schistosoma japonicum*) or vasculature surrounding the urinary bladder (*Schistosoma haematobium*) (*Wilson, 2009*; *Georgi et al., 1986*). The sexually mature male and female worms pair up and generate a prodigious number of eggs, depositing hundreds to thousands of eggs per female worm per day (*Cheever, 1969*; *Cheever et al., 1994*; *Barlow and Meleney, 1949*). Eggs lodge within the perito-neal organs, evoking a massive Th2 response which cues granuloma formation and consequent pathology in affected organs (reviewed in [*Cheever, 1969*]). In the case of urogenital schistosomiasis (*S. haematobium* infection) eggs are deposited in the bladder mucosa and urinary tract, resulting in hematuria and increased rates of squamous-cell carcinoma. In the case of intestinal schistosomiasis (*S. mansoni* and *S. japonicum* infections), eggs are deposited in the liver and hepatic portal system leading to periportal fibrosis, pulmonary hypertension and ascites.

Broad spectrum chemotherapy of schistosomiasis relies upon praziquantel (PZQ), which has been in clinical use for nearly 40 years (*Colley et al., 2014*; *King and Dangerfield-Cha, 2008*; *Andrews et al., 1983*). Mass drug administration (MDA) programs will require ~250 million tablets

**eLife digest** More than 200 million people worldwide are infected with parasitic worms that cause the disease schistosomiasis. Most cases occur in sub-Saharan Africa. Long-term infections can damage organs, and children who are affected may suffer delayed growth and learning difficulties. Despite its significant health and economic impact, schistosomiasis is still considered a 'neglected' tropical disease. This means there has not been adequate investment into developing new treatments or cures.

A drug called praziquantel is currently the only treatment for schistosomiasis. However, the drug has unpleasant side effects, cannot cure all infected individuals, and there is a concern that worms may develop resistance to its effects. This means there is an urgent need to develop new therapies. One possible approach would be to develop drugs that interfere with the worm's ability to move.

Chan et al. screened thousands of existing chemicals for interactions with a protein that is known to control how the worms move. A drug called ergotamine, which is currently used to treat migraines, strongly interacted with the protein. Treating infected mice with ergotamine eliminated the parasites and reduced the organ damage caused by the infection. Praziquantel also reduced the number of parasites in the mice but it did not prevent organ damage.

The results presented by Chan et al. show that a single drug can interact with targets in both the worm and the animals it infects. Searching for drugs that have this dual effect may help to develop more effective treatments for schistosomiasis and other diseases caused by parasites. Ergotamine itself is unlikely to be used to treat people for schistosomiasis because of the side effects produced when using it repeatedly. However, these findings will help researchers identify and develop safer drugs with similar benefits.

DOI: https://doi.org/10.7554/eLife.35755.002

of PZQ per year for at risk populations (*Osakunor et al., 2018*). The widespread distribution of schistosomiasis, coupled with observations that PZQ can be subcurative in areas of high intensity of infection and transmission (*Danso-Appiah and De Vlas, 2002*; *King et al., 2011*) raise concerns that treatment resistant parasites may emerge (*Fallon and Doenhoff, 1994*; *Fallon et al., 1995*; *Ismail et al., 1996*; *Wang et al., 2012*). Therefore, it is important to identify new flatworm drug targets and lead compounds to expand the arsenal of drugs available to treat schistosomiasis and other parasitic flatworm infections currently resolved through PZQ administration.

In this regard, targeting parasite neuromuscular physiology has proven to be a highly successful anthelmintic approach (*Geary et al., 1992*). In schistosomes, serotonin (5-HT) controls motor function and a specific serotonergic G-protein coupled receptor (GPCR) mediating this effect has been implicated by both RNAi (*Patocka et al., 2014*) and drug screening (*Chan et al., 2016a*, *2016b*). Here, we cloned full-length 5-HT receptor sequences from each of three major species causing human infection worldwide and expressed these targets in a high-throughput capable assay enabling screening of thousands of compounds from natural product libraries. The dataset identified anti-schistosomal chemotypes that conveyed anti-parasitic efficacy. Most importantly these activities led to the discovery that the ergot alkaloid ergotamine ameliorated both infection and the pathological sequelae of infection. These properties highlight an opportunity to combat schistosome infections and their pathological impact on the host using single agents that coalesce deleterious actions on parasites with beneficial activities on host responses.

## Results

### Pharmacological profiling of a schistosome 5-HTR

Full-length sequences for a serotoninergic GPCR (5-HTR) were cloned from the three major schistosome species causing infections worldwide; *Schistosoma mansoni* (Sm.5HTR$_L$[*Chan et al., 2016b*]), *S. haematobium* (Sh.5HTR) and *S. japonicum* (Sj.5HTR). The three 5HTRs shared high amino acid similarity (84–94%), and clustered with 5-HT$_7$ receptors from other organisms (*Figure 1—figure supplements 1* and *2*). In mammalian HEK293 cells, GFP-tagged variants of each 5-HTR localized at the cell surface (*Figure 1A*), enabling functional profiling. To measure signaling output, GPCR activity was

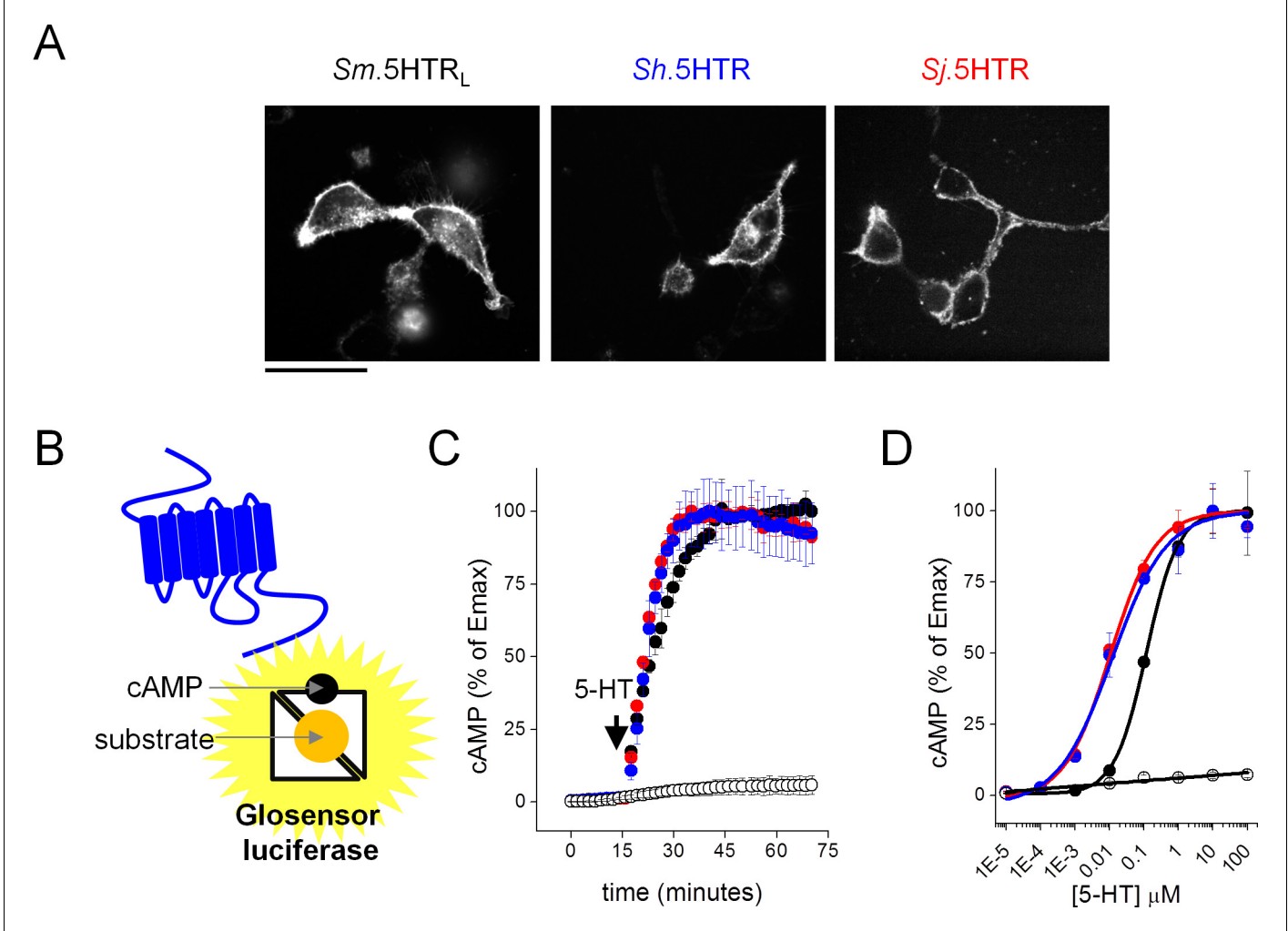

**Figure 1.** Functional expression of *Schistosoma* serotonin receptors. (**A**) Confocal images showing cell surface expression in mammalian HEK293 cells of 5-HT receptors cloned from *S. mansoni* (*Sm*.5HTR$_L$), *S. haematobium* (*Sh*.5HTR) and *S. japonicum* (*Sj*.5HTR) localized by COOH-terminally tagged eGFP. Scalebar, 50 μm. (**B**) Schematic of luminescent cAMP sensor bioassay. cAMP generated by schistosome 5-HTRs (blue) binds the engineered GloSensor luciferase, switching the sensor to a more active conformation resulting in enhanced luminescence output. (**C**) Kinetics of signal following the addition of 5-HT (1 μM) to HEK293 cells co-transfected with luminescent cAMP sensor and individual schistosome 5-HT receptors. Open circles, cells not transfected with 5-HT receptor, colored circles represent measurements in cells transfected with *Sm*.5HTR$_L$ (black), (*Sh*.5HTR (blue) and *Sj*.5HTR (red). (**D**) Serotonin dose-response curves for each of the three receptors (colored circles) and HEK293 cells expressing the cAMP sensor alone (open circles). Data reflect mean ± standard error of at least 3 biological replicates.

DOI: https://doi.org/10.7554/eLife.35755.003

The following figure supplements are available for figure 1:

**Figure supplement 1.** Schistosome 5-HT receptor protein alignment.
DOI: https://doi.org/10.7554/eLife.35755.004

**Figure supplement 2.** Cladogram of schistosome 5-HT receptors.
DOI: https://doi.org/10.7554/eLife.35755.005

**Figure supplement 3.** Performance of the Sm.5HTR$_L$ luminescent cAMP reporter assay.
DOI: https://doi.org/10.7554/eLife.35755.006

assessed using a luciferase based cAMP reporter (GloSensor 22F, Promega) engineered to undergo a conformational change on cAMP binding that increases luciferase activity (*Figure 1B*). The choice of this reporter was guided by prior evidence demonstrating coupling of the flatworm SER-7 GPCRs to G$_s$ (*Patocka et al., 2014*; *Chan et al., 2016b*, *2016c*) and the ability of this approach to resolve real time kinetics of cAMP generation with high sensitivity in intact cells. The 5-HTRs from each

individual schistosome species were transiently transfected into a stable HEK293 cell line expressing the cAMP reporter and luminescence values monitored during addition of 5-HT. Each of the GPCRs responded to the addition of 5-HT with a rapid elevation in luminescence values (~70–110 fold increase, *Figure 1C*) with sensitivity in the low nanomolar range for Sj.5HTR and Sh.5HTR ($EC_{50}$ = 3.0 ± 0.3, 3.6 ± 0.3 nM) and 76 ± 1 nM for Sm.5HTR$_L$ (*Figure 1D*). To facilitate high-through-put drug screening of these targets, dual stable cell lines expressing both the cAMP reporter and individual 5-HTR clones were derived and individual lines evaluated based on performance in a 384-well format. Conditions were iterated to deliver robust resolution of the signals elicited by 5-HT ('agonist screen', Z' score >0.7, signal window >10) and the known inhibitory action of bromocriptine ('antagonist screen', Z' score >0.6, *Figure 1—figure supplement 3*), a previously identified antagonist (*Chan et al., 2016b*).

Having optimized these methods, we proceeded to screen the *S. mansoni* 5HTR$_L$ against a compilation of natural product libraries (4288 compounds, *Supplementary file 1*). The screening protocol consisted of an 'agonist screen' (addition of compound alone, 10 µM) followed by an 'antagonist screen' (inhibition of response to 5-HT, 1 µM) in the same cells for every compound. The workflow, and criteria for putative 'hit' designation, from the primary screen are shown in *Figure 2A*. An example of the data profile generated for a putative agonist 'hit' and an antagonist 'hit' is shown in *Figure 2B*.

Compilation of the screening dataset is shown in *Figure 2C*, where the fold-increase ('agonist screen') or decrease (relative to 5-HT, 'antagonist screen') in luminescence values are plotted for individual compounds. From the whole dataset (4288 ligands), 92 putative hits were identified that exceeded threshold criteria (*Figure 2C*). These compounds were then counter-screened against the parental HEK293 reporter cell line lacking any schistosome 5-HTR (*Figure 2D*). This was necessary to exclude false positive hits, for example compounds that stimulated cAMP production through endogenous HEK293 targets (GPCRs, direct adenylate cyclase activators), or compounds that non-specifically inhibited cAMP production. Counter-screening triaged 40 of the original 92 hits, invalidating compounds that either stimulated cAMP production or inhibited forskolin-evoked cAMP generation in cells lacking Sm.5HTR$_L$. Overall, the screening dataset yielded 52 compounds (12 agonists and 40 antagonists at Sm.5HTR$_L$, ~1.2% hit rate) for subsequent investigation. Chemotype analysis of the 'hit' compounds (*Figure 2E*) revealed two main groups of favored structures. The first core group of ligands were benzylisoquinolines with aporphine or protoberberine cores. These compounds were exclusively antagonists. The second group of compounds contained an indole ring system, either bicyclic tryptamines, or ergot alkaloids with a four ring ergoline core. This second group of compounds comprised both agonist and antagonist ligands. This range of efficacies underpinned prioritization of these ligands, and ergot alkaloids as the predominant chemotype, for further analysis of the structure-activity relationship at the parasitic 5-HTRs.

## Structural determinates of activity at schistosome 5-HTRs

Ergot alkaloids are a diverse group of compounds with a long history of therapeutic use. Their interaction with mammalian bioaminergic GPCRs encompasses a range of activities and kinetic profiles (*Knight et al., 2009*; *Wacker et al., 2017*). Based on the results from the natural product screen, an expanded series of ergot alkaloid ligands was sourced for structure-activity screening ('SAR by commerce'). Complete dose-response relationships were performed for twenty compounds containing the tetracyclic ergoline skeleton (5 compounds from the primary screen plus 15 commercially sourced ligands) against all three schistosome 5.HTRs (Sm.5HTR$_L$, Sh.5HTR and Sj.5HTR).

Ergot alkaloid interactions with the schistosome 5-HTRs spanned a range of effects. To summarize these actions, agonist efficacy (relative to peak levels of 5-HT evoked cAMP generation) and antagonist potency (pKi) were represented as heat-maps where increasingly intense coloration represents higher efficacy (*Figure 3A*) and potency (*Figure 3B*). Several features of this dataset merit comment. First, these representations convey a shared responsiveness across the three 5-HTRs to tested ligands, suggesting a conservation of pharmacological profile between the 5-HTR orthologs from the different schistosome species. This is important if any putative lead compounds are to be broad spectrum, that is effective against each of the major infective strains. Second, the data evidence a broad spectrum of potencies and efficacies within the ergot alkaloid series evidencing full agonists (for example, compound '1' ergotamine, '2' dihydroergotamine), partial agonists (compound '3', lisuride), antagonists (compounds '16' terguride, '19', bromocriptine, '20' 2-bromo-LSD) as well as

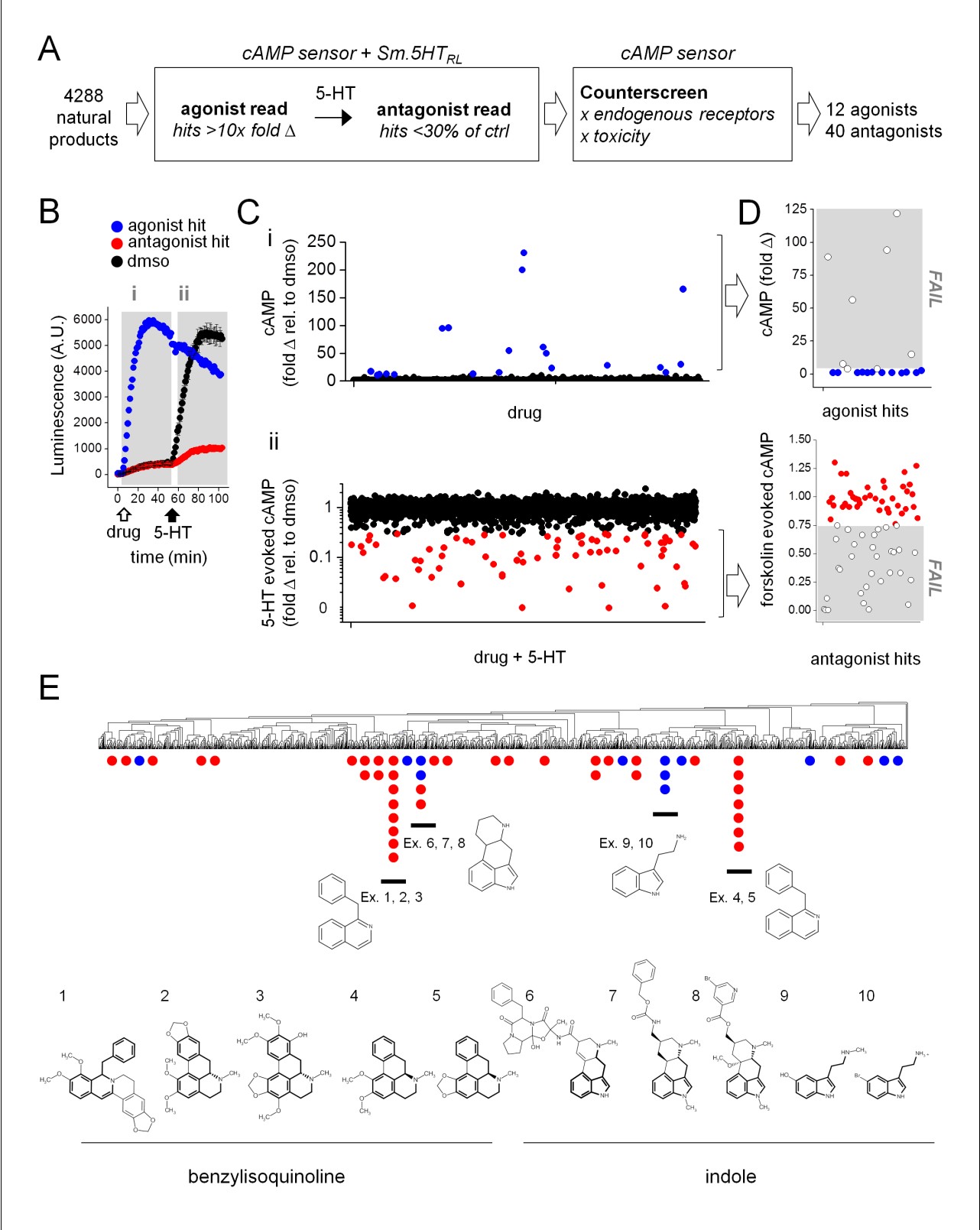

**Figure 2.** High-throughput screen of natural product libraries against *Sm*.5HTR$_L$. (**A**) Assay workflow for drug screen. Compounds were screened against HEK293 cells stably expressing a cAMP sensor and *Sm*.5HTR$_L$, followed by counter-screening hits against a cell line lacking *Sm*.5HTR$_L$. (**B**) Kinetic readout of luminescence from cAMP biosensor following addition of (i) test compound (10 µM, open arrow; agonist hits increase luminescence), followed by (ii) the addition of 5-HT (1 µM, solid arrow; antagonist hits decrease luminescence relative to controls). (**C**) Scatter plot of compounds

*Figure 2 continued on next page*

*Figure 2 continued*

assaying for (i) agonists and (ii) antagonists. Putative hits surpassing threshold are shown in color (blue, agonists; red, antagonists). (D) Counter-screen of putative hit compounds from (C) against cell lines lacking 5-HT receptor to exclude compounds with off-target increases in cAMP (top) or compounds that reduce forskolin (25 μM) evoked cAMP signals (bottom). (E) Hits clustered into compound classes by structure. Groups contain common ring systems such as benzylisoquinoline (ex. 1. ST059293, 2. Nantenine, 3. 785163, 4. Nuciferine, 5. Remerine) or indole (6. Ergotamine, 7. Metergoline, 8. Nicergoline, 9. N-methylserotonin, 10. 5-bromotryptamine) structures. Blue = agonists, red = antagonists.
DOI: https://doi.org/10.7554/eLife.35755.007

closely related 'inactive' compounds (compound '14', α-ergocryptine). Third, marked changes in efficacy and inhibitory potency resulted from minimal structural modifications to the core scaffold. For example, hydrogenation of the D9-10 double bond of lisuride (compound '3') eliminated efficacy to yield the antagonist terguride (compound '16', $IC_{50}$ = 400 ± 50 nM, *Figure 3C*). The partial agonist LSD (compound '6') is converted into a potent antagonist ($IC_{50}$ = 100 ± 20 nM) on bromination at the B2 indol ring position (2-Bromo-LSD, compound '20'). Distinct structural modifications impacting efficacy could be seen in an ergopeptine series (*Figure 3D*), through comparison of the chemical structures of ergotamine (compound '1'), a potent full agonist at each schistosome receptor ($E_{max}$95 ± 4, 96 ± 5 and 114 ± 3% of 5-HT for Sm.5HTR$_L$, Sh.5HTR and Sj.5HTR, respectively), the inactive α-ergocryptine (compound '14') and bromocriptine (compound' 19) which acted as a potent blocker ($IC_{50}$ = 1.6 ± 0.4 μM) of Sm.5HTR$_L$ (*Figure 3D*). Such SAR data underscore the potential for modifying the ergoline scaffold to maximize, or engineer away, efficacy at schistosome 5HTRs (*Figure 3C and D*).

Overall, structure-activity analysis across the three schistosome 5-HTRs identified ergotamine (compound '1') as the most potent, full agonist and 2-Bromo-LSD (compound '20') as the most potent antagonist for further evaluation.

## Validation of Sm.5HTR$_L$ ligands on schistosomes ex vivo

The effects of prioritized compounds from the screen were examined on the movement of adult parasites cultured ex vivo. Male and female worms respond to 5-HT with increased movement (*Figure 4A*). This responsiveness permits the basic screening assay shown in *Figure 4B*, where changes in worm motility are resolved in response to compound addition, and in the presence of 5-HT. For example, 5-HT (200 μM) increased worm movement, and 5-HT stimulation was not observed following preincubation with bromocriptine (1 μM, *Figure 4B*). Compounds that exhibited agonist activity at each of the three 5-HTRs in vitro also stimulated worm movement ex vivo (*Figure 4C*). While the rank order of potency varied slightly from the in vitro screen, an increase in potency of ergot alkaloids relative to 5-HT was evident ($EC_{50}$ for 5-HT = 51.6 ± 3.0 μM, $EC_{50}$ = 97.8 ± 31.6 nM for ergotamine, $EC_{50}$ = 1.4 ± 0.5 μM for LSD). As expected, the potent antagonist BOL-148 (*Figure 3C*) blocked 5-HT evoked hyperactivity (*Figure 4D*). BOL-148 displayed ~10 fold higher potency than bromocriptine consistent with the results from the in vitro screen ($IC_{50}$ = 28.8 ± 23.1 nM for BOL-148 versus $IC_{50}$ = 331 ± 271 nM for bromocriptine, *Figure 4D*).

## Antischistosomal action of Sm.5HTR$_L$ ligands in vivo

Do these schistosome 5-HTR ligands display anthelmintic activity in vivo? As processes such as parasite feeding, mate pairing and movement within the host vasculature depend upon coordinated neuromuscular function, exposure to ligands that stimulate hyperactivity (5-HTR agonists) or cause paralysis (5-HTR antagonists) may promote dysfunction and parasite clearance.

One indicator of acute drug efficacy that may prove indicative of anthelmintic activity is a shift in location of the parasites from the mesenteric vessels to the liver (the 'hepatic shift'), where worms are subsequently targeted for elimination by the host immune system (*Buttle and Khayyal, 1962*; *Pellegrino et al., 1977*; *Mehlhorn et al., 1981*). To examine the ability of the prioritized 5-HT ligands to cause this hepatic shift, mice harboring a mature schistosome infection (42 days p.i.) were given 5-HTR ligands by intraperitoneal injection and sacrificed three hours later to assess parasite distribution (*Figure 5A*). Numbers of worms residing within the mesenteries, portal vein or liver were assessed. Following treatment with PZQ (100 mg/kg), most of the worms were found within the liver (68.8 ± 14.1% of recovered parasites, *Figure 5B*), whereas worms were infrequently found

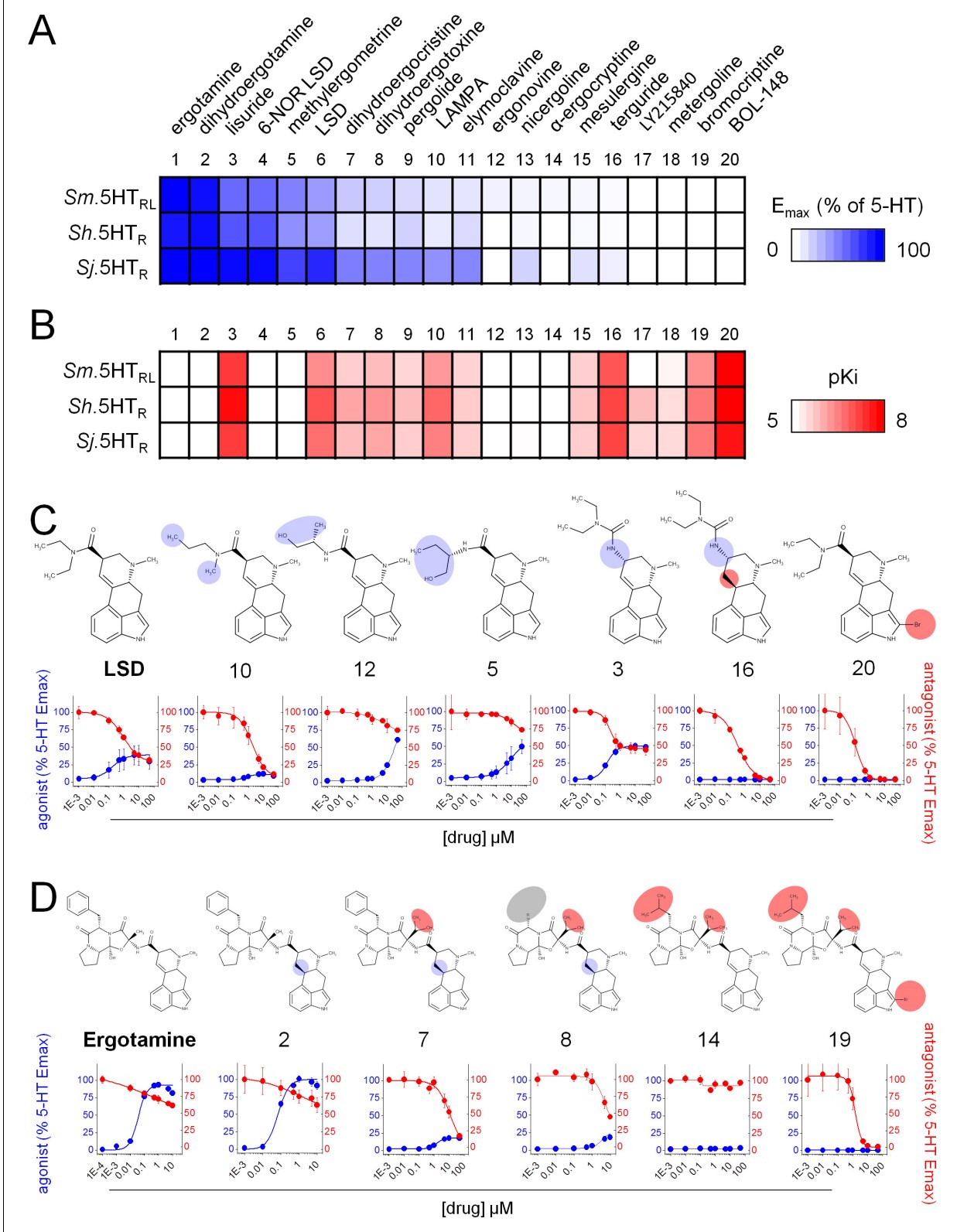

**Figure 3.** Structure activity relationship of ergot alkaloid compounds. Ergot alkaloids display a spectrum of activity against *Schistosoma* 5-HT receptors ranging from (A) agonists (expressed as maximal cAMP response relative to 5-HT) to (B) antagonists (estimated receptor affinity expressed as pKi). (C) Structure activity relationship within a lysergic acid amide series. Derivatives of the partial agonist LSD displayed modified activity following modification of the diethylamide moieties (*Danso-Appiah and De Vlas, 2002*; *Fallon and Doenhoff, 1994*; *Cheever, 1969*; *Wilson, 2009*) or the

*Figure 3 continued on next page*

*Figure 3 continued*

ergoline ring system (*Geary et al., 1992*; *Chan et al., 2016c*). (D) Structure activity relationship within a ergopeptine chemical series. Derivatives of the full agonist ergotamine with altered activity following modification of the tripeptide structure (*Barlow and Meleney, 1949*; *Andrews et al., 1983*; *Ismail et al., 1996*) or the ergoline ring system (*Chan et al., 2016b*). Compound numbering reflects rank order of efficacy across the schistosome 5-HTRs: 1. Ergotamine; 2. Dihydroergotamine; 3. Lisuride; 4. 6-NOR LSD; 5. Methylergometrine; 6. LSD; 7. Dihydroergocristine; 8. Dihydroergotoxine; 9. Pergolide; 10. LAMPA; 11. Elmoclavine; 12. Ergonovine; 13. Nicergoline; 14. α-ergocryptine; 15. Mesulergine; 16. Terguride; 17. LY215840; 18. Metergoline; 19. Bromocriptine; 20. BOL-148. Data represent mean ± standard error of at least three biological replicates.
DOI: https://doi.org/10.7554/eLife.35755.008

within the liver in mice injected with vehicle control (9.7 ± 8.5% of recovered parasites). The performance of a selection of 5-HTR agonists or antagonists was then benchmarked relative to these standards.

Treatment with 5-HTR agonists elicited a hepatic shift (*Figure 5B*). However, treatment with 5-HTR antagonists did not alter worm distribution (*Figure 5B*). Furthermore, the extent of the hepatic shift observed in mice treated with 5-HTR agonists correlated with the measured efficacy of these agonists at the parasite 5.HTR (*Figure 5B*). Full agonists promoted the greatest shift: for example, ergotamine treatment of mice yielded a similar percentage of worms in the liver (75.1 ± 16.9%) as seen in mice treated with PZQ. Partial agonists promoted a lesser response, and 5-HTR antagonists were ineffective. For example, neither bromocriptine (2.0 ± 2.8% of worms in liver) or BOL-148 (5.2 ± 2.3% of worms in liver) caused a hepatic shift of worms (*Figure 5B*) despite the effectiveness of these compounds at arresting worm movement ex vivo (*Figure 4D*). Based on these data, we prioritized treatment of infected mice with ergotamine to assess amelioration of chronic schistosomiasis.

## In vivo efficacy of ergotamine

Mice were injected with vehicle control (DMSO), PZQ (50 mg/kg) or ergotamine (60 mg/kg) twice daily for one week starting 42 days post infection (p.i.). Control animals succumbed to infection (~8–10 weeks p.i.) over a timeframe consistent with the onset of egg laying by sexually mature parasites as expected (*Figure 6A*, [*Cheever, 1969*]). By 15 weeks p.i., the majority of control mice were dead. Death rates were significantly reduced in ergotamine-treated mice (16.5% versus 68.9% mortality in control cohort, log-rank test p<0.001, hazard ratio 5.02). PZQ-treated mice survived through 15 weeks with no observed lethality (*Figure 6A*).

As ergotamine increased survival of infected mice, we assessed various metrics of infection (worm burden, egg number, hepatosplenomegaly) immediately following ergotamine treatment (42–49 days p.i). First, ergotamine treatment significantly reduced worm burden from 28 ± 8 worm pairs per mouse to 9 ± 6 worm pairs per mouse (~65% reduction, p<0.002; *Figure 6B*). Next, egg counts were performed. Eggs deposited within the mesenteric system progress through the intestinal mucosa for excretion from the host, thereby propagating the parasite life cycle (*Pellegrino and Faria, 1965*). Therefore, it is possible to determine whether drug treatment impacted parasite egg laying by counting eggs in sections of the large and small intestines. Control mice contained large numbers of viable eggs in both the small and large intestine. However, treatment with either ergotamine or PZQ resulted in a considerable reduction (96.3 ± 1.2 and 99.9 ± 0.1%, respectively) in intestinal egg burden (*Figure 6C*). We also assessed whether serotonergic manipulation influenced egg laying in worms cultured ex vivo (*Figure 6—figure supplement 1A*). Egg laying was decreased following treatment with ergotamine (1 μM, 54.2 ± 12.6% reduction), serotonin (100 μM, 82.4 ± 2.2% reduction) or forskolin (100 μM, 61.6 ± 12.6% reduction) (*Figure 6—figure supplement 1B&C*). Eggs laid by ergotamine treated worms were often small and deformed, indicating abnormal development (*Figure 6—figure supplement 1D*). We conclude ergotamine was highly effective at blocking parasite egg production and development.

Next, liver and spleen sizes were measured. A consequence of the enormous production of schistosome eggs is a host Th2 wave (*Grzych et al., 1991*) that drives granuloma formation and fibrosis in affected peritoneal organs (*Pearce and MacDonald, 2002*). Hepatosplenomegaly is a hallmark of chronic schistosomiasis and splenomegaly and liver fibrosis are observed in the murine model (*Fallon, 2000*). This is reflected by comparison of the weights of uninfected mouse spleens (0.12 ± 0.02 g) and the spleens of infected littermates (~6 fold enlargement, weighing 0.76 ± 0.20 g at 49 days p.i), or increases in liver mass (~2 fold enlargement from 2.0 ± 0.3 g in uninfected mice to 4.8 ± 0.8 g

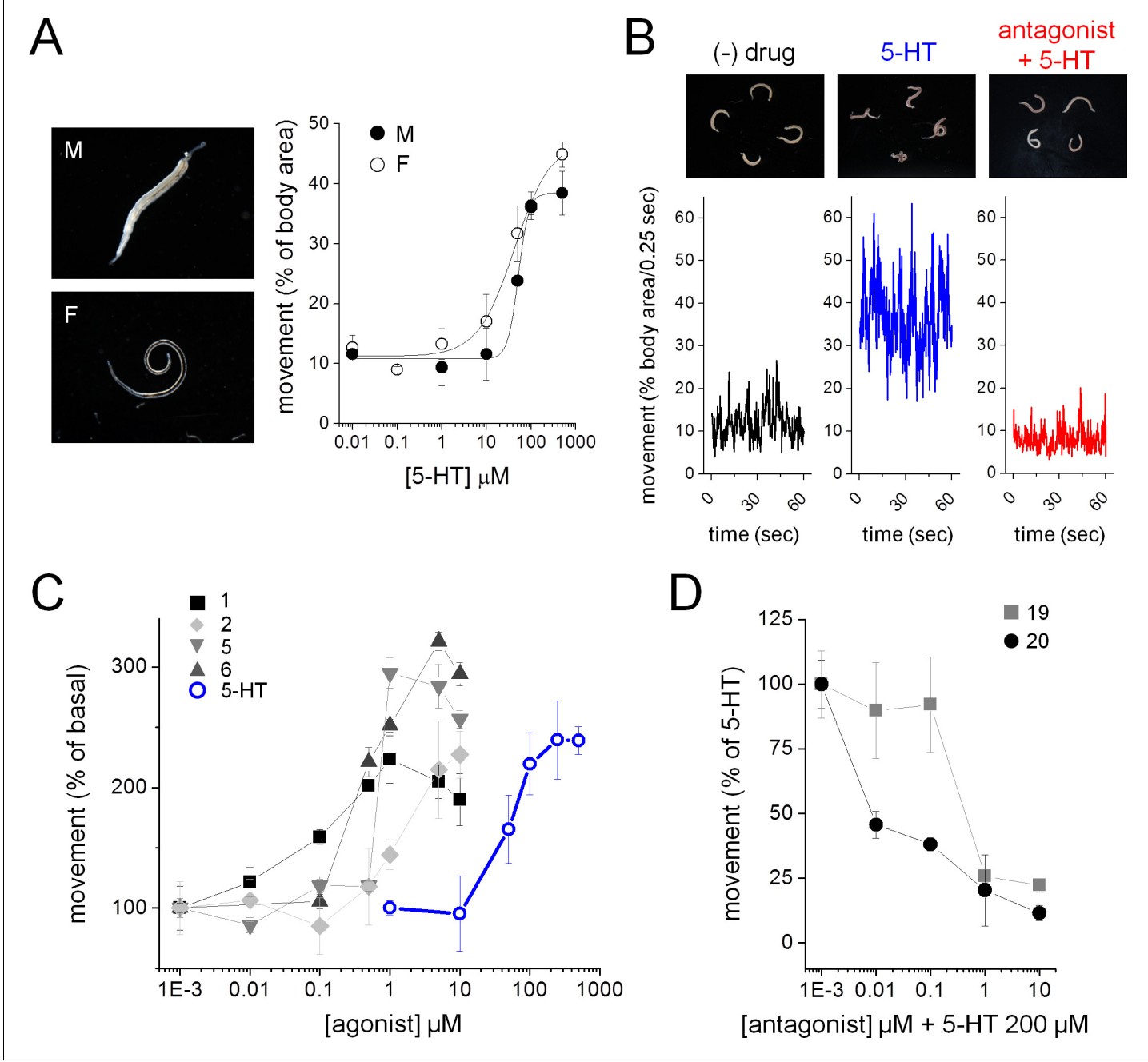

**Figure 4.** Ergot alkaloid action against adult *S. mansoni* parasites ex vivo. (**A**) Serotonin induced increase in movement of *S. mansoni* adult male (M, solid circles) and female (F, open circles) parasites. (**B**) Adult *S. mansoni* movement in the absence of drug (black trace), following 5-HT addition (200 μM, blue), or exposure to a *Sm*.5HTR$_L$ antagonist (BOL-148, 1 μM) and subsequent 5-HT (200 μM, red) addition. (**C**) Dose-response curves for *Sm*.5HTR$_L$ agonist evoked stimulation of adult schistosome movement. Ergot alkaloids = solid symbols, 5-HT = open blue circles. (**D**) Dose-response curves for *Sm*.5HTR$_L$ antagonist inhibition of 5-HT (200 μM) evoked movement. Worm movement data (**A, C–D**) reflect mean ± standard error of the mean for at least 3 biological replicates.

DOI: https://doi.org/10.7554/eLife.35755.009

in infected mice, **Figure 6D**). PZQ treatment (one week, starting at 42 days p.i) did not prevent spleen and liver enlargement (**Figure 6D**). There was no difference in the weights of the liver and spleen between PZQ-treated and vehicle treated infected mice (**Figure 6E and F**). This is likely due to the fact that, while PZQ is highly effective at eliminating worms at six weeks infection and beyond

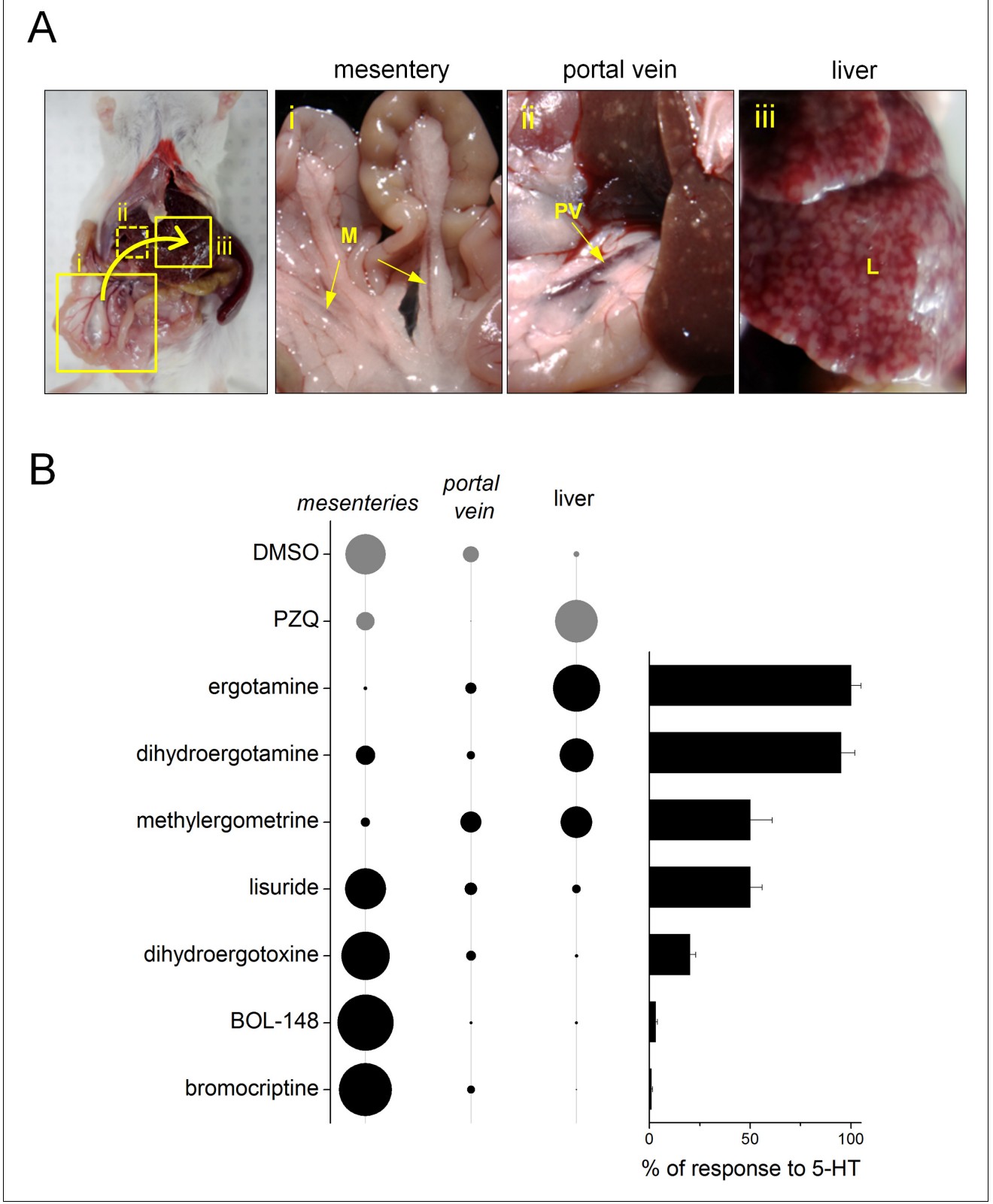

**Figure 5.** Ergot alkaloid agonists evoke an acute hepatic shift of parasites in vivo. (**A**) Location of mouse vasculature dissected after drug treatment to resolve worm location. Adult parasites normally reside in the mesenteric vasculature (i), but following anthelmintic treatment shift to the portal vein (ii) and liver (iii). (**B**) Left, quantification of percent worm burden found in either the mesenteries, portal vein or liver following treatment with DMSO

*Figure 5 continued on next page*

*Figure 5 continued*

(negative control), praziquantel (100 mg/kg, positive control) and various ergot alkaloid compounds. Diameter of circle represents proportionality of distribution. Right, bar chart represents efficacy of the same series of compounds at *Sm.*5HTR$_L$.

DOI: https://doi.org/10.7554/eLife.35755.010

(*Sabah et al., 1986*; *Keiser et al., 2009*), sexual maturation and egg laying commence as early as 5 weeks post infection (*Sabah et al., 1986*), meaning that eggs deposited prior to treatment initiation still evoke an immune response which PZQ exposure does not ameliorate. In contrast to PZQ, ergotamine exposure prevented hepatosplenomegaly. Ergotamine reduced the infection-promoted increase in spleen mass by 55.8 ± 14.0% and liver mass by 52.0 ± 14.2% relative to vehicle treated controls (*Figure 6E and F*). Similarly, liver sections from ergotamine-treated infections displayed decreased granuloma formation and fibrosis when investigated by various staining methods (*Figure 6G* and *Figure 6—figure supplement 2*). Movat's stain and Masson's trichrome showed decreased levels of collagen deposition surrounding granulomas (*Figure 6G,II–III*). Aldehyde fuchsin stain showed elastin surrounding the granulomas of DMSO and PZQ treated mice, while the livers of uninfected or ergotamine treated mice exhibited little staining (*Figure 6G,IV*). Oil red O staining revealed lipid stores present in the livers of uninfected mice that were absent in infected control livers. Treatment with ergotamine, but not PZQ, preserved regions of Oil red O positive staining - albeit at lower levels than healthy livers (*Figure 6G,V*). Pro-apoptotic cleaved caspase-3 was present in cells within granulomas of all three infected livers (DMSO control, ergotamine and PZQ treated – *Figure 6G,VI*).

Finally, we tested whether ergotamine displayed efficacy as a treatment against immature infections, where PZQ is ineffective (*Sabah et al., 1986*). Both 5-HT and ergotamine stimulated movement of juvenile worms ex vivo (*Figure 6—figure supplement 3A and B*). In vivo, however, early dosing with ergotamine prior to egg laying (3–4 weeks post infection) did not resolve infections, mirroring the lack of effect of PZQ (*Figure 6—figure supplement 3C*).

In conclusion, these data show that ergotamine was highly effective at promoting survival in mice with mature infections (*Figure 6A*) through reductions in worm (*Figure 6B*) and egg number (*Figure 6C*). However - unlike PZQ - ergotamine was effective at attenuating hepatosplenomegaly associated with the chronic pathology of schistosomiasis.

## Ergotamine impairs activation of hepatic stellate cells

How does ergotamine treatment protect against hepatosplenomegaly resulting from schistosome infection? Both ergotamine and PZQ clear parasites, but unlike ergotamine, PZQ-treated mice did not show a noticeable difference in liver and spleen size relative to control infections (*Figure 6*). As ergotamine is a human therapeutic with affinity for various host GPCRs (*O'Connor and Roth, 2005*), in addition to high affinity for schistosome 5-HT receptors (*Figure 3D*), engagement of host signaling pathways likely underpins these protective effects.

To investigate mechanisms contributing to the differential in vivo effects of these drugs, we performed RNA-Seq analyses on the livers and spleens of schistosome-infected mice following drug treatment (PZQ, ergotamine) over a window 6 to 7 weeks post-infection (*Figure 7A*). Livers and spleens of infected mice showed widespread changes in gene expression relative to uninfected littermates; of the 24,421 transcripts with mapped sequencing reads, 9684 transcripts were differentially expressed in infected livers and 7325 transcripts were differentially expressed in infected spleens. It is well established that the host pathology of schistosomiasis infection is driven by an initial Th1 response to the worms themselves followed by a sustained Th2 response that drives egg-induced fibrosis (reviewed in [*Pearce and MacDonald, 2002*; *Wynn et al., 2004*]). Such a gene expression signature has been observed in numerous studies on livers or spleens of infected mice (*Sandler et al., 2003*; *Burke et al., 2010*; *Gobert et al., 2015*; *Perry et al., 2011*) and is borne out here through our data comparing infected and uninfected controls. Most components of Th1 and Th2 signaling pathways were significantly changed in both infected livers and spleens relative to uninfected littermates (*Figure 7B and C*).

Drug treatment with either PZQ or ergotamine attenuated transcriptional changes associated with infection. The identity of transcripts changing with drug treatment (either with PZQ or with

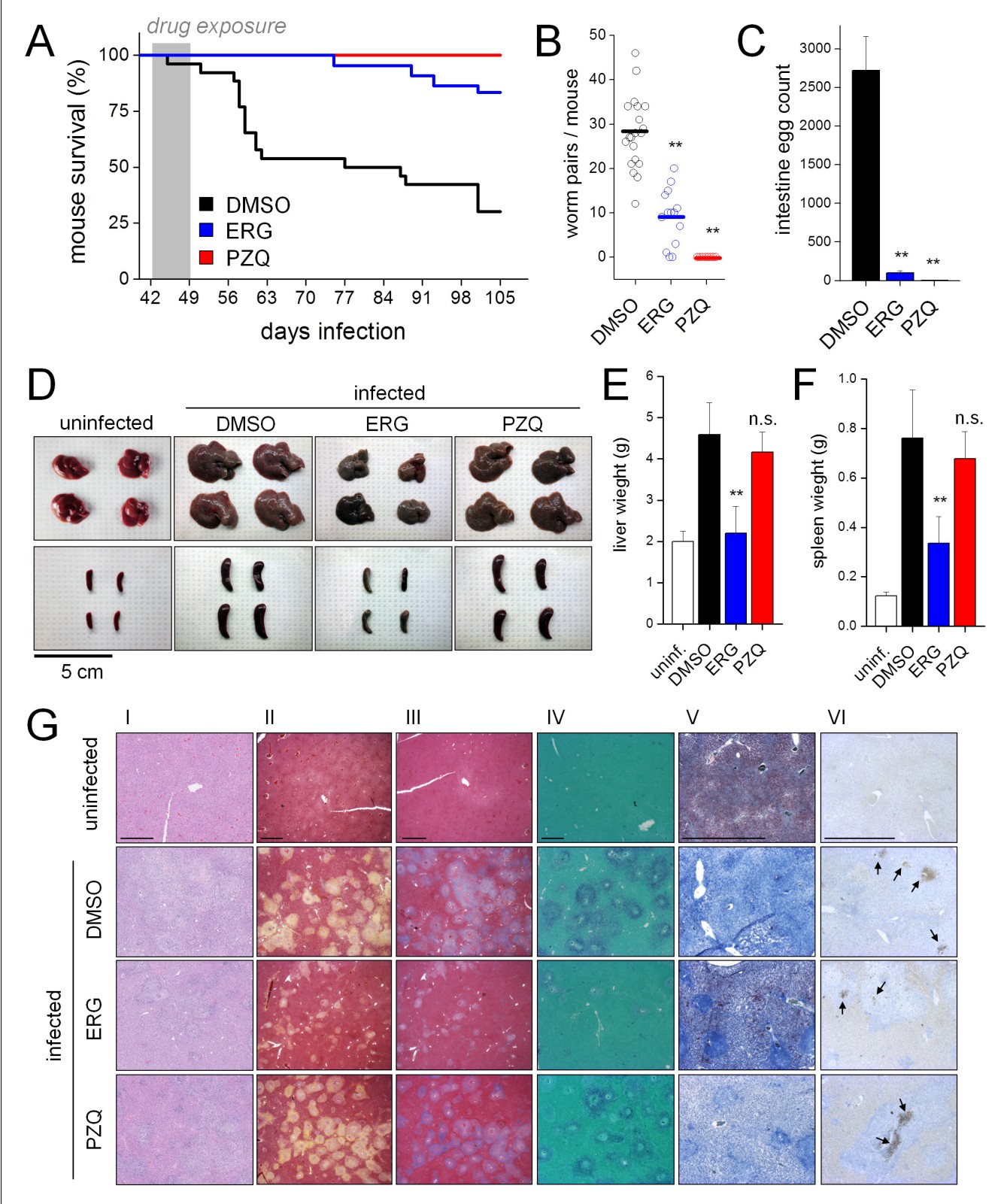

**Figure 6.** Ergotamine resolves infections and mitigates pathology of schistosomiasis in vivo. (A) Survival of mice infected with *S. mansoni* and treated for one week (days 42–49 post infection) with either vehicle control (DMSO. n = 26 mice), ergotamine (ERG, 60 mg/kg twice daily. n = 22 mice) or praziquantel (PZQ, 50 mg/kg daily. n = 10 mice). (B) Worm burden of mice treated as in A and sacrificed 49 days post-infection. (C) Intestinal egg counts of mice sacrificed in B. (D) Representative livers and spleens from uninfected mice and infected mice exposed to vehicle (DMSO), ergotamine

*Figure 6 continued on next page*

*Figure 6 continued*

(ERG) or praziquantel (PZQ). Quantification of hepatomegaly (**E**) and splenomegaly (**F**) in uninfected mice (open columns) and infected mice following drug treatment (solid columns). (**G**) Histology of liver sections control uninfected and infected (DMSO) mice, as well as drug treated (ERG and PZQ) infected animals (scale = 1 mm). Liver histology using (I) H and E staining, (II) Movat's stain (collagen, yellow), (III) Masson's trichrome stain (collagen, blue), (IV) aldehyde fuchsin stain (elastin, purple), (V) Oil red O stain (lipid, red), and (VI) caspase-3 activation (brown puncta, arrows). Higher magnification images of egg granulomas for each staining condition are shown in *Figure 6—figure supplement 2*.

DOI: https://doi.org/10.7554/eLife.35755.011

The following figure supplements are available for figure 6:

**Figure supplement 1.** Ergotamine inhibition of *S. mansoni* egg laying during ex vivo culture.

DOI: https://doi.org/10.7554/eLife.35755.012

**Figure supplement 2.** Liver egg granulomas in control and drug treated mice.

DOI: https://doi.org/10.7554/eLife.35755.013

**Figure supplement 3.** Effects of ergotamine on immature *S. mansoni*.

DOI: https://doi.org/10.7554/eLife.35755.014

**Figure supplement 4.** Lack of hepatic microvesicular steatosis in drug treated livers.

DOI: https://doi.org/10.7554/eLife.35755.015

ergotamine treatment versus control infections) largely overlapped with those changing with infection (uninfected animals versus control infections) but as expected drug-treatment was associated with a decrease of transcripts up-regulated in control infections (relative to uninfected animals) or an increase of transcripts down-regulated in control infections (relative to uninfected animals, *Figure 7D*). Specifically, PZQ or ergotamine treatment decreased expression of pathways associated with host immune response to either parasites (acute phase response signaling, Th1 and Th2 pathway) or bacteria entering the circulation with the breakdown of the intestinal epithelium that occurs with heavy parasite egg production ultimately leading to host death (LPS/IL-1 mediated inhibition of RXR function [*Feingold et al., 2004*; *Wang et al., 2005*]). Expression of pathways associated with drug metabolism (xenobiotic metabolism signaling) and liver function (hepatic cholestasis) increased with both treatments (*Figure 7D*). These observations that the PZQ and ergotamine cohorts shared similar transcriptional patterns relative to control infection is consistent with both treatments resolving infection and parasite egg production (*Figure 6B and C*).

Next, we sought to filter the transcriptomic dataset to identify transcriptional changes unique to ergotamine treatment (*Figure 7E*). This involved selecting transcripts that (i) displayed significant changes between uninfected and control infection cohorts, and (ii) were differentially expressed between control infection and ergotamine-treated cohorts, and (iii) showed little change between control infection and praziquantel treated cohorts (*Figure 7E*). Ingenuity pathway analysis using this triple-filtered criteria identified a cohort of transcripts uniquely associated with ergotamine, but not PZQ, treatment. These transcripts were tagged for involvement in hepatic fibrosis and hepatic stellate cell (HSC) activation (p-value 0.00015, Fisher's exact test). Ergotamine treatment down-regulated gene products associated with HSC activation to control infections, while PZQ treatment had little to no effect on the same suite of transcripts. HSCs represent a quiescent, non-proliferative cell population in normal liver but once activated during injury or disease, drive fibrotic responses by transforming into proliferative, contractile myofibroblasts that deposit extracellular matrix components (*Tsuchida and Friedman, 2017*). Differentially expressed transcripts in ergotamine-treated samples (*Supplementary file 4*) included known markers of HSC activation responsible for extracellular matrix production (*COL1A1, COL1A2, COL3A1*), transcripts associated with HSC contraction (*CACNA1E, CACNA1H*), migration (*IGFBP3, RANTES*), survival (*IGFBP5, GAS6*) and transdifferentiation into myofibroblasts (*MMP13, GLI1, PAK1*) (*Figure 7F and G*). Ligands that control HSC activation and survival were also differentially expressed (*PDGFB, IL13RA2, BMP4, IL-4, SAA*).

Therefore, ergotamine and PZQ were both effective at promoting survival (*Figure 6A*) through reductions in worm (*Figure 6B*) and egg number (*Figure 6C*), as well as reversing many transcriptional signatures that drive host pathological outcomes in response to parasite infection (*Figure 7D*). However, unlike PZQ, ergotamine was also effective at attenuating the hepatosplenomegaly associated with the chronic pathology of schistosomiasis (*Figure 6D–G*), due to decreased activation of profibrotic HSCs (*Figure 7E–G*).

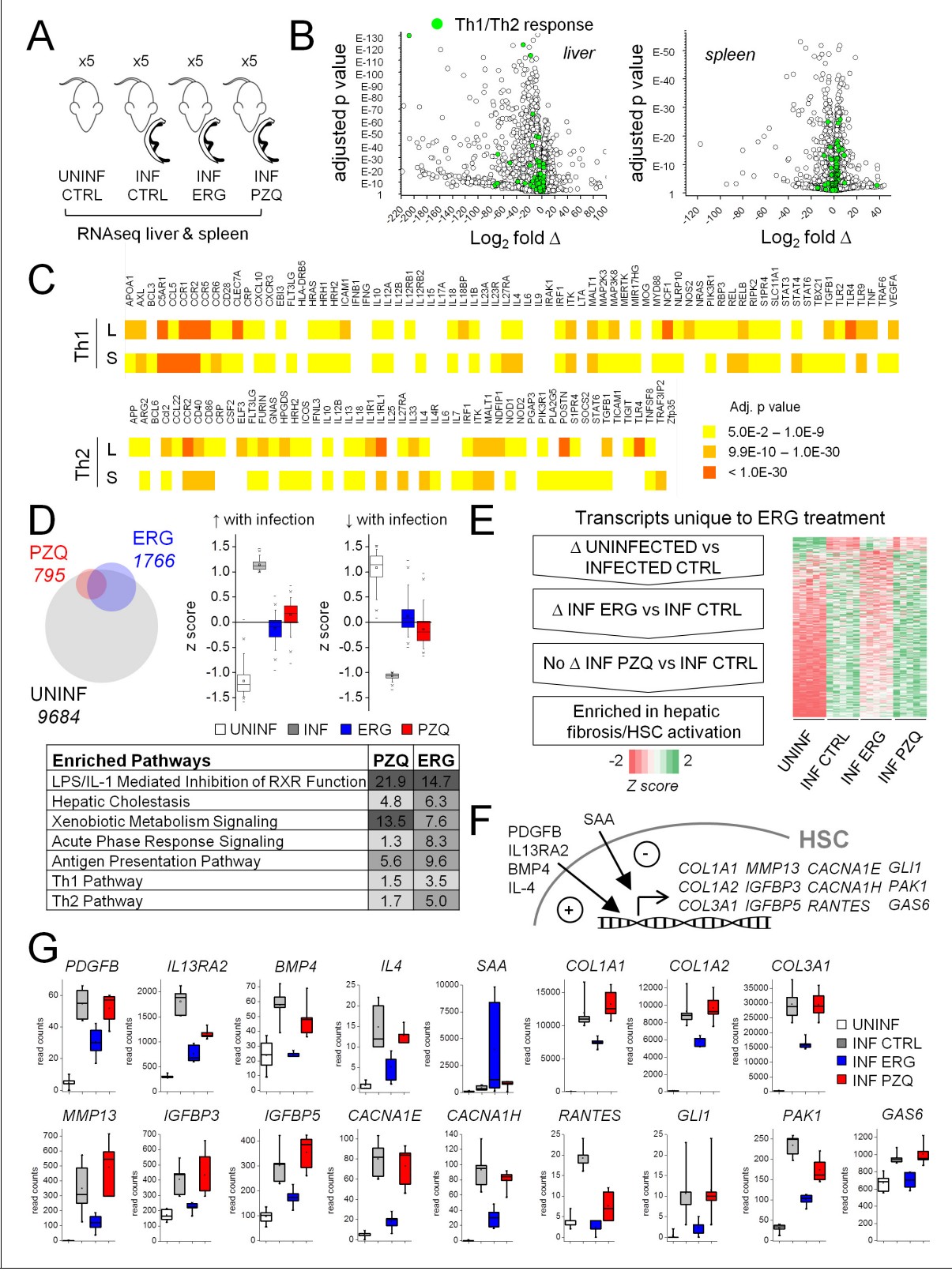

**Figure 7.** Effect of ergotamine on immune response and hepatic stellate cell activation. (**A**) Schematic of schistosome infection and drug treatment cohorts analyzed. Livers and spleens were harvested from infected mice treated with vehicle control, ergotamine (ERG 60 mg/kg, twice daily for 7 days) or praziquantel (PZQ 50 mg/kg, once daily for 5 days), as well as uninfected littermate controls. Five mice were used per cohort. (**B**) Volcano plots showing transcriptional changes of the liver and spleens from infected mice compared to uninfected littermates. Green shading highlights gene

*Figure 7 continued on next page*

*Figure 7 continued*

products involved in Th1/Th2 cell signaling. (C) Heat map of liver (L) and spleen (S) changes in Th1 and Th2 signaling pathways - color intensity corresponds to significance (FDR adjusted p-values). (D–G) Effects of anthelmintics on hepatic gene expression of schistosome infected mice. (D) Top left - differentially expressed transcripts comparing the livers of infected controls to uninfected littermates (grey), PZQ-treated animals (red) and ERG-treated animals (blue). Top right - expression of transcripts either up-regulated or down-regulated in infected livers relative to uninfected livers (z score $\geq 1$ or $\leq -1$) in control and drug treated animals. Bottom, transcripts differentially expressed between PZQ-treated or ERG-treated and control infections represent pathways involved in liver function, drug metabolism and immune response. Ingenuity Pathway Analysis enrichment scores reflect -$\log_{10}$ p-value of PZQ and ERG datasets (p-value 0.05 = -$\log_{10}$ 1.3). (E) Filtering of transcripts changing with schistosomiasis infection that are affected by ERG but not PZQ, resulting in an enrichment for gene products involved in hepatic fibrosis/HSC activation. (F) Role of various gene products in HSC activation, and (G) read counts for these transcripts across uninfected (open box), infected (grey box) and ERG-treated (blue box) and PZQ-treated (red box) cohorts. Box plots represent 25–75 percentile, whiskers 5–95 percentile. Inset; bar = median, square = mean. Read count data for liver and spleen samples available in **Supplementary files 2** and **3**.

DOI: https://doi.org/10.7554/eLife.35755.016

## Discussion

Given the importance of serotonergic signaling in parasite physiology, the possibility of serotonergic drugs providing effective antiparasitic agents has long been recognized (*Mansour, 1979*). With a defined molecular target (a prevalent schistosome 5-HTR cloned from each of the main species that cause schistosomiasis) and an optimized functional profiling assay for this GPCR (*Figure 1*), we have now been able to investigate this possibility. Here, we screened thousands of compounds in vitro (*Figure 2*) and interpolated structure-activity studies on promising chemical series (*Figure 3*). These cell-based assays yielded candidates for ex vivo screening against adult parasites (*Figure 4*), and thereby prioritization of the most active compounds for in vivo testing in a mouse model of schistosomiasis (*Figure 5*). This pipeline identified unique properties of ergotamine, an ergot alkaloid first isolated 100 years ago (*Hofmann, 1978*) and FDA-approved medication for migraine, for treating schistosomiasis. Ergotamine increased survival in a murine model of schistosomiasis through reductions in parasite burden and resultant egg load, but most intriguingly resolved inflammatory damage caused by schistosome eggs laid in host tissues, an effect not observed with PZQ under the same conditions (*Figure 6*). These effects likely result from the broad polypharmacology of ergotamine that coalesces a fortuitous combination of effects at both host and parasite GPCRs.

### Action on parasite

Ergotamine is a potent Sm.5HTR$_L$ agonist (*Figure 3*). Sm.5HTR$_L$ and homologs of this receptor in other flatworms, regulate movement (*Patocka et al., 2014*; *Chan et al., 2015*). Sm.5HTR$_L$ is downregulated upon pairing of adult *S. mansoni* females and males, implicating an additional role in sexual maturation (*Lu et al., 2016*). As a G$_s$ coupled receptor, the role of 5-HT acting through cAMP to regulate flatworm metabolism is also relevant (*Mansour, 1984*). These broad physiological roles - motor function, female egg production, metabolism – underscore consideration of Sm.5HTR$_L$ as an attractive antischistosomal target. Screening hits (*Figure 2*), and subsequent structure-activity studies (*Figure 3*) confirmed the ergoline core as a customizable scaffold for ligand optimization at Sm.5HTR$_L$. These analyses provided a toolbox of compounds with varied potencies and efficacies against Sm.5HTR$_L$ in vitro and Sm.5HTR$_L$ agonists and antagonists stimulated and inhibited adult worm motility ex vivo as predicted (*Figure 4*). However in vivo, only agonists of Sm.5HTR$_L$ signaling caused a hepatic shift of worms and mitigated worm burden, Sm.5HTR$_L$ blockers were ineffective (*Figure 5*). This observation was surprising given the presumed beneficial effect of causing potent (BOL-148) or long-lasting (bromocriptine) worm paralysis (*Chan et al., 2016a*, *2016c*), but nevertheless informative for future phenotypic screens to prioritize 'hyperactive' over 'hypoactive' serotoninergic hits. From these analyses, ergotamine emerged as a potent full agonist (*Figure 3D*) that stimulated movement ex vivo (*Figure 4C*) and reduced worm and disease burden in vivo (*Figure 6*).

### Action on host

While ergotamine stimulates schistosome movement through 5.HTRs, the drug also acts on host GPCR targets. Ergotamine is known to display a broad polypharmacology, interacting with numerous bioaminergic GPCRs (*O'Connor and Roth, 2005*). These interactions likely engage physiological responses that mitigate the damage caused by egg laying and possibly contribute to worm

elimination. Acutely, ergotamine acts as a vasoconstrictor, including within the mesenteric vasculature where the adult worms reside (*Mikkelsen et al., 1981*). Disruption of the parasite environment consequent to ergotamine-evoked mechanical changes and alterations of blood flow within the hepatic portal system could drive parasites from the mesenteries to the liver for elimination. This action would mirror recent data demonstrating R-PZQ, which also causes a hepatic shift, acts a human 5-HT$_{2B}$ receptor partial agonist that contracts host mesenteric vessels (*Chan et al., 2017*). Chronically, drug treatment with either PZQ or ergotamine improved various aspects of liver function (hepatic cholestasis, retinol metabolism, bile acid synthesis, *Figure 7D* and *Supplementary file 4*) relative to untreated infections. Beneficial changes highlighted by RNA-Seq also encompassed protection against intestinal damage (*Cao et al., 2010*). Various pathways involved in response to bacterial infection were down regulated with PZQ and ergotamine treatment relative to control infections (*Supplementary file 4*). This is likely because mice experience a relative infection burden orders of magnitude higher than typical for humans (*Cheever, 1969*) resulting in compromised intestinal integrity and release of microbiota into the circulation as mature parasites commence egg production. These effects likely contribute to the observed lethality post-infection and are shown to be mitigated by both drug treatments in the RNA-Seq analysis.

However, ergotamine additionally provided protection against the spleen and liver enlargement that persisted in PZQ-treated mice (*Figure 6D and E*), suggesting a further action of ergotamine on the host immune response. Analysis of RNA-Seq datasets identified transcriptional changes unique to ergotamine and not PZQ treatment, most notably a decrease in gene products involved in hepatic fibrosis (*Figure 7E–G*). Hepatic stellate cells (HSCs) are key drivers of fibrosis during liver disease. Upon their transformation into myofibroblasts, HSCs initiate active production of extracellular matrix components (collagen, elastin – *Figure 6G,II-IV*) and decrease lipid storage (*Figure 6G,V*, [*Tsuchida and Friedman, 2017*]). HSCs are known to express a portfolio of GPCRs that regulate activation and transformation (*Tsuchida and Friedman, 2017*). These include several serotonergic GPCRs (5HT$_{1B}$, 5HT$_{1F}$, 5HT$_{2A}$, 5HT$_{2B}$, 5HT$_7$, [*Ruddell et al., 2006*]). It is tantalizing that the known polypharmacological signature of ergotamine (*O'Connor and Roth, 2005*) closely matches the serotoninergic GPCR expression profile in HSCs (*Ruddell et al., 2006*). This may imply a cell-autonomous action of ergotamine on HSCs, although further insight would be needed to distinguish this possibility from non cell-autonomous mechanisms given that serotonin impacts immune responses and liver regeneration in a multiplicity of ways (*Herr et al., 2017*; *Lesurtel et al., 2006*; *Ebrahimkhani et al., 2011*). For example, ergotamine could impair HSCs indirectly by inhibiting macrophage polarization (*de las Casas-Engel et al., 2013*) or immune mediators such as TNF-α and ICAM-1 (54). ICAM-1 is found at elevated levels within egg-evoked granulomas (*Ritter and McKerrow, 1996*) and TNF-α drives the host immune response to schistosome eggs (*Amiri et al., 1992*; *Leptak and McKerrow, 1997*). Prior studies on schistosome infection of immune compromised mice have shown that T-cell deprived mice exhibit hepatic microvesicular steatosis, since granulomas also have a beneficial function of facilitating excretion of eggs from the mesenteries and sequestering toxic egg products (reviewed in [*Hams et al., 2013*; *Doenhoff et al., 1986*]). Ergotamine treatment impaired the granulomatous response to eggs (*Figure 6G*, *Figure 6—figure supplement 2*), but increased microvascular damage was not observed (*Figure 6—figure supplement 4*). Additional experiments will be needed to investigate stellate cell responsiveness and whether ergotamine is effective against other parasitic diseases currently treated with PZQ.

While the polypharmacology of ergotamine evidences the principle of combining host and parasite responses for therapeutic effect, it is not itself a likely candidate for repurposing. Repeated dosing of ergotamine as an antimigraine therapy is associated with known side effects: it is contraindicated in pregnancy and poorly suited for the large pediatric population susceptible to schistosomiasis. For future studies, analogs with improved pharmacokinetic profile and bioavailability will need to be evaluated for clinical potential as safe and effective agents. Indeed, for treatment of parasitic infections, a single bolus dose may provide more flexibility and avoid cumulative side effects associated with repeated dosing. Our limited structural-activity studies to date (e.g. *Figure 3*) on schistosome GPCRs do suggest modifications of the core ergot alkaloid scaffold can generate analogs with a range of potencies and efficacy.

In conclusion, ergotamine is fortuitously suited to engage pathways in both host and parasite that are simultaneously deleterious to the blood fluke but beneficial to the infected host in terms of worm clearance and mitigation of damage caused by eggs. Rather than prioritizing 'selective'

antiparasitic therapies with no host effects, these data demonstrate merit in evaluating agents with broader polypharmacology across parasite and host as a route to novel discovering effective treatments for parasitic disease.

# Materials and methods

## Key resources table

| Reagent type (species) or resource | Designation | Source or reference | Identifiers | Additional information |
|---|---|---|---|---|
| Cell line | HEK293 cells | Company | ATCC; CRL-1573 | |
| Biological sample (*Schistosoma mansoni*) | *Schistosoma mansoni* | Stock Center | BEI: NR-34792 | Biodefense and Emerging Infections Research Resources Repository (BEI Resources) |
| Biological sample (*Schistosoma japonicum*) | *Schistosoma japonicum* | Stock Center | BEI: NR-45100; BEI:NR-45101 | Biodefense and Emerging Infections Research Resources Repository (BEI Resources) |
| Biological sample (*Schistosoma hematobium*) | *Schistosoma hematobium* | Stock Center | BEI: NR-45102; BEI:NR-45103 | Biodefense and Emerging Infections Research Resources Repository (BEI Resources) |
| Recombinant DNA reagent | Sm.5HTRL | Data repository | GenBank accession KX150867 | |
| Recombinant DNA reagent | Sj.5HTR | Data repository | GenBank accession number MG813904 | |
| Recombinant DNA reagent | Sh.5HTR | Data repository | GenBank accession number MG813903 | |
| Commercial assay or kit | GloSensor 22F cAMP sensor | Company | Promega; E2301 | |
| Commercial assay or kit | D-luciferin sodium salt | Company | Gold Biotechnology; LUCK-100 | |
| Commercial assay or kit | Cleaved caspase −3 | Company | Biocare; CP-229C | |
| Chemical compound, drug | NCI Natural Products Set IV | Other | | National Institutes of Health, NCI Development Therapeutics Program (DTP) |
| Chemical compound, drug | Natural Product Library | Company | TimTec: NPL-800 | |
| Chemical compound, drug | Natural Derivatives Library | Company | TimTec: NDL-3000 | |
| Chemical compound, drug | LSD | Other | NIDA; 7315–006 | National Institute on Drug Abuse (NIDA) |
| Chemical compound, drug | BOL-148 | Other | NIDA; 7315–010 | National Institute on Drug Abuse (NIDA) |
| Chemical compound, drug | PZQ - praziquantel | Company | Sigma Aldrich: P4668-1G | |
| Chemical compound, drug | ergotamine | Company | Sigma Aldrich: 1241506–150 MG | |
| Chemical compound, drug | dihydroergotamine | Company | Sigma Aldrich: D1952000 | |
| Chemical compound, drug | bromocriptine | Company | Sigma Aldrich: B2134-100MG | |
| Chemical compound, drug | IBMX | Company | Sigma Aldrich: I5879-100MG | |
| Chemical compound, drug | LAMPA | Company | Cerilliant: L-004–1 ML | |
| Chemical compound, drug | Lisuride | Company | Tocris: 4052 | |

*Continued on next page*

*Continued*

| Reagent type (species) or resource | Designation | Source or reference | Identifiers | Additional information |
|---|---|---|---|---|
| Chemical compound, drug | dihydroergotoxine | Company | Tocris: 0474 | |
| Chemical compound, drug | methylergometrine | Company | Tocris: 0549 | |
| Chemical compound, drug | forskolin | Company | Cell Signaling Technology: 3828 | |
| Software, algorithm | HISAT2 | PMID: 25751142 | | |
| Software, algorithm | edgeR | PMID: 19910308 | | |
| Software, algorithm | Ingenuity Pathway Analysis | Company | QIAGEN Bioinformatics: IPA | |
| Software, algorithm | DAVID Bioinformatics Resources 6.8 | Other | | Laboratory of Human Retrovirology and Immunoinformatics (LHRI) |

## Molecular cloning

The sequence for Sm.5HTR$_L$ (GenBank accession KX150867 [*Chan et al., 2016a*]) was used to BLAST the *S. haematobium* and *S. japonicum* genomes for putative homologs. The resulting *S. haematobium* (NCBI Reference Sequence XM_012944163 and XM_012944164) and *S. japonicum* (GenBank FN332592.1) hits were used as templates to clone out full-length mRNA sequences by 5' and 3' RACE (Marathon cDNA Amplification Kit, Clontech) using *S. haematobium* and *S. japonicum* total RNA provided by the Schistosomiasis Resource Center. Finalized sequences have been deposited with NCBI (Sh.5HTR = GenBank accession number MG813903, Sj.5HTR = GenBank accession number MG813904). Plasmids for heterologous expression were generated by codon optimization of the coding sequence for mammalian expression and subcloning into either pcDNA3.1(-) (between *Not*I and *Eco*RI) or pEGFP-N3 (*Eco*RI).

## Cell lines and culture

HEK293 cells (ATCC CRL-1573, an authenticated cell line validated by STR profiling) were cultured in growth media consisting of DMEM supplemented with GlutaMAX (Gibco cat # 10566016) + 10% heat inactivated fetal bovine serum and penicillin-streptomycin (100 units/mL, ThermoFisher). Stable cell lines for the pGloSensor-22F cAMP Plasmid (Promega) and schistosome 5HTRs in pcDNA3.1(-) were selected with hygromycin (200 µg/mL) and G418 (400 µg/mL). The correct identity of each stably expressed sequence was verified by isolating total RNA from each stable line (TRIzol Reagent, Ambion) and amplifying the sequence of interest with gene specific primers flanking the 5' and 3' regions of the coding sequence (SuperScript III One-Step RT-PCR System, Invitrogen). PCR products were ligated into a TA cloning vector (pGEM-T Easy, Promega) and verified by Sanger sequencing.

## cAMP luminescence assays

HEK293 lines stably expressing both the GloSensor 22F cAMP sensor and Sm.5HTR$_L$ were cultured in growth media supplemented with 10% dialyzed FBS. For cAMP assays in 96 well format, cells cultured in T-75 flasks were trypsinized (TrypLE Express, Gibco) and plated in solid white 96 well plates (Costar cat # 3917) the day prior to assay at a density of $2.5 \times 10^4$ cells/well. The following day media was removed and replaced with assay buffer consisting of HBSS buffered with HEPES (20 mM, pH 7.4) + BSA (0.1% w/v) and D-luciferin sodium salt (1 mg/mL, Gold Biotechnology). Plates were equilibrated at room temperature for 1 hr, at which point 3-Isobutyl-1-methylxanthine (IBMX 200 µM, Sigma Aldrich) was added. Plates were equilibrated a further 30 min prior to test compound addition. Test compounds were added and luminescence was read for 45 min on a GloMax-Multi Detection System plate reader (Promega) to screen for agonists, after which plates were removed, 5-HT was added to each well at a concentration determined to achieve a maximal response for each stable cell line (Sm.5HTR$_L$ = 500 nM, Sj.5HTR = 500 nM, Sh.5HTR = 60 nM), and plates were read a second time to screen for antagonist activity. Readings performed in 384 well format were modified so that cells were assayed in suspension. Cells were grown to 70% confluence in T-75 flasks,

trypsinized and pelleted (300 RCF x 5 min). Media was removed and cells were resuspended in 25 mL assay buffer supplemented with IBMX (200 µM). A 96 channel semi-automated pipet (Eppendorf epMotion 96) delivered 55 µL of cell suspension per well (5000 cells/well) into solid white 384 plates (Corning 3574). After one hour of equilibration, test compounds were screened at a fixed concentration of 10 µM, and control wells (64 wells per plate) were treated with vehicle alone (1% DMSO in HBSS), providing an internal negative control reference. Following 60 min incubation at ambient temperature, plates were read for agonist activity, defined as a >10 fold increase in luminescence relative to control, DMSO treated wells on the same plate. Serotonin (1 µM) was then added to each well, and after 60 min the plate was read again to identify compounds with antagonist activity, defined as compounds that inhibiting 5-HT evoked signal by >90% relative to control wells. The initial high-throughput screen in 384 well format was performed as a single replicate, with all 'hits' meeting agonist or antagonist thresholds re-screened in 96 well format (technical quadruplicates performed for three biological replicates). Counter screening of primary hits against the parental HEK293 cell line expressing the GloSensor 22F cAMP sensor and lacking Sm.5HTR$_L$ was performed to eliminate compounds stimulating cAMP (>3 fold change in basal luminescence) or inhibiting forskolin evoked cAMP production (>25% inhibition relative to DMSO controls).

## Reagents and chemicals

A complete list of chemical vendors, catalog numbers and SMILES is contained in *Supplementary file 1*. Natural product libraries were sourced from the National Institutes of Health (NCI Natural Products Set IV) and TimTec (Natural Product Library [NPL-800] and Natural Derivatives Library [NDL-3000]). LSD and BOL-148 were sourced from the National Institute on Drug Abuse (NIDA). The following compounds were sourced from Sigma Aldrich: praziquantel (PZQ, P4668-1G), ergotamine tartrate (1241506–150 MG), dihydroergotamine tartrate (D1952000), bromocriptine (B2134-100MG). LAMPA was sourced from Cerilliant (L-004–1 ML). Lisuride maleate (4052), dihydroergotoxine mesylate (0474) and methylergometrine maleate (0549) were sourced from Tocris.

## Ethics statement

All animal experiments followed ethical regulations approved by the Medical College of Wisconsin and Iowa State IACUC committees and additionally reviewed in the context of funding by the National Institutes of Health (NIAID).

## Ex vivo schistosome assays

Female Swiss Webster mice infected with *S. mansoni* cercariae (NMRI strain) were sacrificed 49 days post infection by $CO_2$ euthanasia. Adult schistosomes were recovered by dissection of the mesenteric vasculature. Harvested schistosomes were washed in RPMI 1640 Medium with GlutaMAX + 5% heat inactivated FBS (Gibco) and Penicillin-Streptomycin (100 units/mL).

For movement assays, worms were cultured 37°C and 5% $CO_2$ in vented 100 × 25 mm petri dishes (ThermoFisher cat # 4031) containing 50 mL of media and used 1–3 days after harvesting. Prior to assessing worm movement, male worms were transferred to a six well dish (4–5 individual worms per well) containing 3 mL drug solution in RPMI 1640 supplemented with HEPES (25 mM) and FBS (5%). Videos were recorded using a Zeiss Discovery v20 stereomicroscope and a QiCAM 12-bit cooled color CCD camera controlled by Metamorph imaging software (version 7.8.1.0). 1 min recordings were acquired at 4 frames per second, saved as a. TIFF stack, and movement was analyzed using ImageJ software as described in (*Chan et al., 2016a*, *2016b*). Data represents mean ± standard error for $\geq3$ independent experiments. EC$_{50}$ and IC$_{50}$ values are reported ± 95% confidence interval of fitted curves.

For egg laying assays, adult schistosomes were transferred to 24 well plates the day after harvesting from mice (5 pairs of male and female worms in 2 mL media/well). Eggs were counted daily using a stereomicroscope, after which worms were transferred to a new well with fresh drug-containing media. Egg counts were recorded for 5 days, and data processed as the mean number of eggs laid per worm pair per day. Measurements of egg dimensions were quantified using the 'analyze>measure' function in ImageJ to record the length and width of individual eggs.

## In vivo schistosome drug screening

Female Swiss Webster mice were exposed to 200 *s. mansoni* cercariae (NMRI strain) at between 4–6 weeks old. For hepatic shift assays, infections matured to 7 weeks, at which point mice were given test compounds by intraperitoneal injection and euthanized 3 hr later. Compounds were solubilized in 50 µL DMSO, and diluted in 200 µL 5% w/v Trappsol (Cyclodextrin Technologies Development, THPB-p-31g) in saline (NaCl 0.9%) solution. Immediately after being euthanized, mice were dissected to separate the liver from the portal vein, so that the number of worms recovered from the portal vein, mesenteric vasculature and liver could be recorded for each mouse. Hepatic shift assays were performed on at least 3 mice per drug treatment. Assays testing the curative efficacy of compounds against schistosome infections were performed on mice at 3 weeks post infection (immature parasites) and 6 weeks post infection (mature parasites). Drugs were dosed as follows. Ergotamine (60 mg/kg) was solubilized in 50 µL DMSO and diluted in 200 µL Trappsol-saline and delivered by intraperitoneal injection twice a day for one week. Praziquantel (50 mg/kg) was similarly solubilized and injected intraperitoneally once a day for one week. The negative control cohort was given twice daily injections of DMSO (50 µL) added to 200 µL Trappsol-saline solution. Mice were weighed and euthanized at 49 days post infection. Worms were harvested and counted as described for the hepatic shift assay. Spleen and livers were weighed, and a segment of intestine was excised from the most distal region of the rectum to 10 cm above the cecum. The small and large intestines were separated by cutting immediately above and below the cecum. Each was cut lengthwise to expose the lumen and thoroughly washed in NaCl (1.2%) to remove excrement. Intestines were laid flat with the intestinal mucosa facing upwards and clamped between two glass plates to allow visual inspection of egg morphology and number using a stereo microscope. Scored egg counts reflect 'viable' eggs containing a developing embryo or a mature miracidium, while empty egg shells and granulomas were not included (*Pellegrino and Faria, 1965*; *Mati and Melo, 2013*).

## Liver histology and immunohistochemistry

Mouse livers were removed immediately after animals were euthanized, rinsed in ice cold PBS, and fixed in 10% neutral buffered formalin. For H and E stain, Movat's stain, Masson's trichrome, and aldehyde fuchsin staining, samples were embedded in paraffin and sectioned at 4 microns (Microm HM355S). For Oil red staining, livers were equilibrated in freezing medium (10% w/v and then 30% w/v sucrose solution in PBS) prior to cryosectioning (Leica CM1850 UV Cryostat). If paraffin embedded, slides were deparaffinized, hydrated, and the following staining procedures were then performed: *H and E stain,* slides were stained with Harris hematoxylin and eosin working solution (Poly Scientific). *Movat's stain,* Alcian blue (1% w/v), hematoxylin (10%), crocein scarlet –acid fuchsin solution, and alcoholic saffron (6.5% w/v). *Masson's trichrome,* slides were pre-treated with Bouin's solution and successively stained with hematoxylin, Biebrich-Scarlet-Acid Fuchsin and Aniline Blue solution. *Aldehyde Fuchsin.* slides were stained with aldehyde fuchsin and counterstained with fast green. *Oil red O staining,* after cryosectioning onto Superfrost slides (Fisherbrand), samples were stained with Oil Red (0.3% w/v in 60% isopropanol) followed by hematoxylin counterstain. *Cleaved Caspase-3 staining,* slides were treated with Leica BOND Epitope Retrieval Solution 2 (20 min), peroxidase-blocking solution (Dako), avidin/biotin blocking kit (Vecgtor) and background sniper blocking reagent (Biocare) per manufacturer's instructions. This was followed by Cleaved Caspase-3 primary antibody incubation (Biocare, CP-229C) at 1:100 dilution for 90 min at room temperature, 3 × 1 min washes (Lieca BondTM Wash Solution), and incubation in biotinylated secondary antibody (Jackson Immuno, 771-066-152) at 1:500 dilution for 30 min, 3 × 1 min washes, and finally incubation in Streptavidin HRP (Dako, P0397) at 1:300 dilution. After washing (3 × 1 min), staining was visualized using DAB + Substrate Chromogen System (Dako, K3468). As a negative control, samples were treated as stated above but without primary antibody.

## Transcriptome sequencing

Mice were infected and treated with drugs as described above. After drug treatment from weeks 6–7 post infection, mice were euthanized and livers and spleens were immediately removed and placed on ice. For liver samples, a section of the larger, right lobe was excised and homogenized in TRIzol Reagent (Invitrogen). For spleen samples, the entire organ was homogenized in TRIzol. Samples were stored in −80°C until processed. Total RNA was extracted from TRIzol according to

manufacturer's protocols and quantified with RiboGreen RNA Assay Kit. Libraries were generated using the Clontech StrandedRNA Pico Mammalian kit and sequenced using the Illumina HiSeq 2500 system (high-output mode, 50 bp paired-end reads) at a depth of approximately 20 million reads/sample. Trimmed reads were mapped to the mouse genome (mm10/GRCm38) using HISAT2. Expression was quantified using featureCounts (read counts) and cuffquant (FPKM). EdgeR was used to identify differentially expressed genes (tagwise dispersion model, adjusted p-value<0.05) which were processed using Ingenuity Pathway Analysis (Qiagen) and DAVID functional annotation tool (version 6.8). RNA-Seq data has been deposited in the NCBI SRA database under accession number SRP131511.

## Statistics

Metrics for evaluating cell based screening assays include coefficient of variation (CV%), signal window (SW), Z factor and Z' factor. Equations for each metric are included in the legend of *Figure 1—figure supplement 3*. Sample sizes for in vivo mouse experiments measuring worm burden were chosen to exceed minimum sample sizes determined by power calculations assuming a difference in means of 60%, standard deviation of 20%, power of 0.9 ($\beta = 0.1$) and significance criterion ($\alpha$) of 0.05. Sample size for in vivo mouse survival analysis experiments also assumed a significance criterion ($\alpha$) of 0.05 and power of 0.9 ($\beta = 0.1$), with half of the mice being treated with experimental drug and half with vehicle control and a hazard ratio of 5. RNA-Seq data was analyzed for differential expression in EdgeR which generated FDR adjusted p-values using the Benjamini-Hochberg method - transcripts with p-values under 0.05 were considered differentially expressed.

## Acknowledgements

We thank the NIDA Drug Supply Program for providing reagents (LSD, 2-Bromo-LSD). Schistosome-infected mice were provided by the NIAID Schistosomiasis Resource Center at the Biomedical Research Institute (Rockville, MD) through NIH-NIAID Contract HHSN272201000005I for distribution via BEI Resources.

## Additional information

### Funding

| Funder | Grant reference number | Author |
| --- | --- | --- |
| National Institutes of Health | R21AI25821 | Jonathan S Marchant |
| National Institutes of Health | R21AI130642 | Jonathan S Marchant |
| National Institutes of Health | R01GM088790 | Jonathan S Marchant |
| National Institutes of Health | F32AI124598 | John D Chan |

The funders had no role in study design, data collection and interpretation, or the decision to submit the work for publication.

### Author contributions

John D Chan, Conceptualization, Formal analysis, Funding acquisition, Investigation, Visualization, Writing—original draft, Writing—review and editing; Timothy A Day, Project administration, Writing—review and editing; Jonathan S Marchant, Conceptualization, Supervision, Funding acquisition, Visualization, Writing—original draft, Project administration, Writing—review and editing

### Author ORCIDs

John D Chan http://orcid.org/0000-0003-4986-972X
Timothy A Day http://orcid.org/0000-0002-9037-6540
Jonathan S Marchant http://orcid.org/0000-0001-6592-0877

## Ethics

Animal experimentation: All animal experiments followed ethical regulations approved by the Medical College of Wisconsin IACUC committee (AUA00006079) and additionally reviewed in the context of extramural funding by the National Institutes of Health (NIAID).

## Decision letter and Author response

Decision letter https://doi.org/10.7554/eLife.35755.025
Author response https://doi.org/10.7554/eLife.35755.026

## Additional files

### Supplementary files

• Supplementary file 1. Natural product library screening data. (Sheet 1) Vendor, chemical identification (SMILES) and primary screening data for all natural product compounds screened against Sm.5HTR$_L$. (Sheet 2) Counter screening data for agonist primary hits in cells expressing the 22F cAMP GloSensor and lacking Sm.5HTR$_L$. (Sheet 3) Counter screening data for antagonist primary hits in cells expressing the 22F cAMP GloSensor and lacking Sm.5HTR$_L$. NPS = NCI Natural Products Set IV, NPL = TimTec Natural Product Library, NDL = TimTec Natural Derivatives Library.
DOI: https://doi.org/10.7554/eLife.35755.017

• Supplementary file 2. RNA-Seq data of drug treated and control infected mouse livers. Read counts for uninfected control, infected control, infected ergotamine-treated and infected praziquantel treated liver samples.
DOI: https://doi.org/10.7554/eLife.35755.018

• Supplementary file 3. RNA-Seq data of drug treated and control infected mouse spleens. Read counts for uninfected control, infected control, infected ergotamine-treated and infected praziquantel treated spleen samples.
DOI: https://doi.org/10.7554/eLife.35755.019

• Supplementary file 4. Signaling pathways enriched in drug treated livers relative to control infections. Ingenuity Pathway Analysis (IPA) of transcripts differentially expressed in livers of PZQ (Sheet 1) and ergotamine (Sheet 2) treated mice relative to control infections. IPA generated activation Z-scores are shown if magnitude $\geq 2$. KEGG pathway analysis of transcripts differentially expressed in livers of PZQ (Sheet 3) and ergotamine (Sheet 4) treated mice relative to control infections.
DOI: https://doi.org/10.7554/eLife.35755.020

• Transparent reporting form
DOI: https://doi.org/10.7554/eLife.35755.021

### Data availability

RNA-Seq data has been deposited in the NCBI SRA database under accession number SRP131511.

The following dataset was generated:

| Author(s) | Year | Dataset title | Dataset URL | Database, license, and accessibility information |
|---|---|---|---|---|
| Chan JD, Day TA, Marchant JS | 2018 | RNA-Seq of schistosome infected mus musculus: adult female liver | https://www.ncbi.nlm.nih.gov/sra?term=SRP131511 | Publicly available at the NCBI Sequence Read Archive (accession no: SRP131511) |

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
