## [Decision Letter]

Thank you for submitting your article "Coalescing beneficial host and deleterious antiparasitic actions as an antischistosomal strategy" for consideration by *eLife*. Your article has been reviewed by three peer reviewers, and the evaluation has been overseen by a Reviewing Editor and Wenhui Li as the Senior Editor. The following individuals involved in review of your submission have agreed to reveal their identity: Ray Pierce (Reviewer #2).

The reviewers have discussed the reviews with one another and the Reviewing Editor has drafted this decision to help you prepare a revised submission.

Summary:

In the absence of vaccines and in light of the very limited repertoire of available treatments, the search for new drugs against schistosomiasis is a priority. Chan et al. report here an elegant and creative approach that led to an interesting set of findings on a target-based screen of a *Schistosoma* serotoninergic GPCR receptor. Ergotamine is identified as an agonist hit that negatively influences worm movement. In a mouse model ergotamine decreases mortality of schistosome-infected mice, parasite number, and egg load. At the same time ergotamine treatment also reduces organ pathology in the mouse liver through the engagement of host GPCRs, which suppress the activation of hepatic stellate cells as well as inflammatory damage and fibrosis. Based on high quality data, the authors conclude that ergotamine exerts dual affects by targeting host and parasite GPCRs with a deleterious outcome for the parasite and a beneficial effect for the host.

Essential revisions:

1) Ergotamine is an alkaloid that shares structural similarity with neurotransmitters like dopamine. Acting as an agonist, this drug leads to vasoconstriction and is mainly prescribed for patients suffering from migraines. However, contraindications comprise among others hepatic disease, pregnancy, and the age of patients – such as preschoolers. Since schistosomiasis mainly affects children, including preschoolers, and since it is also a hepatic disease, how safe or how problematic could this drug be for a patient?

According to the Mayo Clinic recommendations (last update: March 01, 2017), 1 – 2 mg are recommended for adults. In case no side effects occur, a second and even a third dose can be taken; however, the doses should be taken at least 30 minutes apart, and should not exceed 3 mg per day. Ergotamine should not be taken by very young children (< 6 years). And from 6 years on, or older, 1 mg is recommended with a maximum of 3 mg a day along with the restriction that this drug should not be taken by children more often than 2 times a week, at least five days apart. Against the background of these facts and the animal data from this study (60 mg/kg twice daily for one week in the mouse model), the low bioavailability in human (2%, when given as tablet; PMCID: PMC1428366), and the short half time of 2 – 3 hours (https://druginfo.nlm.nih.gov/drugportal/rn/113-15-5), how realistic are the chances for ergotamine to become a drug repurposed for schistosomiasis?

The potential problems that may arise with the repurposing of this drug and the extensive clinical testing in the context of treatment for schistosomiasis should be addressed in the Discussion.

2) The antiparasitic effect of ergotamine was accompanied by a marked apparent improvement in the pathological effects of infection on the host. The observed reduction in liver and spleen size and the improvement in liver morphology was found to correlate with an ergotamine-specific effect on hepatic stellate cells, and hence on liver fibrosis. A marked reduction in granuloma size and liver fibrosis is seen in congenitally athymic mice. However, in the case of *S. mansoni* infection, this leads to increased mortality and fewer eggs passed into the feces (Cheever et al., Am J Trop Med Hyg, 1993, 48:496-503). Byram and von Lichtenberg (Am J Trop Med Hyg, 1977, 26:944-56) suggested that the deficient granuloma formation led to failure to sequester toxic egg products, resulting in hepatocellular damage. Although smaller, the livers of ergotamine-treated, infected mice do not seem normal (Figure 5D) and have a darker color, even than control infected or PZQ-treated mice. Further, the picrosirius red-stained histological sections (Figure 5G) could be complemented with other staining methods to determine whether any hepatocyte necrosis is visible. It is important to show that the ergotamine effect on the host response is really beneficial.

3) Do these compounds (e.g., ergotamine) have activity against other schistosome stages (schistosomules, juveniles). Those experiments could be done ex vivo for a start, and subsequently in vivo. This issue is an important one, as juvenile worms (2-4 weeks p.i.) are refractory to PZQ. The issue should at least be discussed. Also do these compounds have activity in motility assays (or, even better, in vivo) against other species of schistosomes? Based on the data in Figure 3, one would expect they would, but it would be nice to know.

4) Why is mouse survival lower in ergotamine-treated than in PZQ-treated infected mice despite the ameliorating effects of ergotamine on pathology.

[Editors' note: further revisions were requested prior to acceptance, as described below.]

Thank you for resubmitting your work entitled "Coalescing beneficial host and deleterious antiparasitic actions as an antischistosomal strategy" for further consideration at *eLife*. Your revised article has been favorably evaluated by Wenhui Li (Senior Editor), a Reviewing Editor, and three reviewers.

The manuscript has been improved but there are some remaining issues that need to be addressed by editing the text before acceptance, as outlined below:

In their revised version, Chan et al. addressed a number of points raised by the reviewers.

With respect to my former comment that: "contraindications comprise among others hepatic disease, pregnancy, and the age of patients – such as preschoolers. Since schistosomiasis mainly affects children, including preschoolers, and since it is also a hepatic disease, how safe or how problematic could this drug be for a patient?" the authors responded that they "acknowledge the side effect profile of ergotamine and agree in the absence of preclinical data that speculation upon the human repurposing potential of ergotamine as a treatment is premature. […] We have therefore removed previous text speculating on the repurposing potential of ergotamine throughout the manuscript text. Additionally, we note: (i) our SAR studies (Figure 3) underscore the potential for manipulation of the core ergot alkaloid pharmacophore to develop analogs with varied efficacy, and potentially improved PK/PD, (ii) treatment of parasitic diseases with a single bolus dose of drug may provide more flexibility than dosing parameters established on the basis of cumulative side effects from repeated drug dosing."

In their marked version, however, it is not possible to follow these deletions since appropriate positions have not been marked as deleted within the text. Please indicate the "removed previous text speculating" clearly. Furthermore, I miss comments within the manuscript with respect to "the potential for manipulation of the core ergot alkaloid pharmacophore to develop analogs with varied efficacy, and potentially improved PK/PD". The chapter "Structural determinates of activity at schistosome 5-HTRs" ends with " ergotamine (compound '1) as the most potent, full agonist and 2-Bromo-LSD (compound '20') as the most potent antagonist for further evaluation." Unfortunately, also the Discussion fails to discuss this point critically. Instead, the Discussion ends with the conclusion that "[…] ergotamine is fortuitously suited to engage pathways in both host and parasite that are simultaneously deleterious to the blood fluke but beneficial to the infected host in terms of worm clearance and mitigation of damage caused by eggs". This is suggestive for its repurposing potential, even without mentioning this word.

In the Abstract the authors mention that "Conventional approaches for antiparasitic drug discovery center upon discovering selective agents that adversely impact parasites with minimal host side effects". Especially the latter seems not applicable against the background of all known contradictions of ergotamine, its low bioavailability, and its limitations for children, which represent the major group of schistosomiasis patients.

Therefore, the authors should critically discuss this point in their Discussion chapter and should mention that "analogs with varied efficacy, and potentially improved PK/PD" based on the ergotamine core are needed to develop a lead that may have the potential to become an effective and safe drug.

---

## [Author Response]

Essential revisions:1) Ergotamine is an alkaloid that shares structural similarity with neurotransmitters like dopamine. Acting as an agonist, this drug leads to vasoconstriction and is mainly prescribed for patients suffering from migraines. However, contraindications comprise among others hepatic disease, pregnancy, and the age of patients – such as preschoolers. Since schistosomiasis mainly affects children, including preschoolers, and since it is also a hepatic disease, how safe or how problematic could this drug be for a patient?According to the Mayo Clinic recommendations (last update: March 01, 2017), 1 – 2 mg are recommended for adults. In case no side effects occur, a second and even a third dose can be taken; however, the doses should be taken at least 30 minutes apart, and should not exceed 3 mg per day. Ergotamine should not be taken by very young children (< 6 years). And from 6 years on, or older, 1 mg is recommended with a maximum of 3 mg a day along with the restriction that this drug should not be taken by children more often than 2 times a week, at least five days apart. Against the background of these facts and the animal data from this study (60 mg/kg twice daily for one week in the mouse model), the low bioavailability in human (2%, when given as tablet; PMCID: PMC1428366), and the short half time of 2 - – 3 hours (https://druginfo.nlm.nih.gov/drugportal/rn/113-15-5), how realistic are the chances for ergotamine to become a drug repurposed for schistosomiasis?The potential problems that may arise with the repurposing of this drug and the extensive clinical testing in the context of treatment for schistosomiasis should be addressed in the Discussion.

We acknowledge the side effect profile of ergotamine and agree in the absence of preclinical data that speculation upon the human repurposing potential of ergotamine as a treatment is premature. Backing up, we believe our contribution in the context of academic drug-discovery is validation of new parasite targets and the discovery of new treatment strategies (drugs to engage both parasite and host responses, rather than just parasite-selective ligands), and would rather highlight our findings with ergotamine in this context. We have therefore removed previous text speculating on the repurposing potential of ergotamine throughout the manuscript text. Additionally, we note: (i) our SAR studies (Figure 3) underscore the potential for manipulation of the core ergot alkaloid pharmacophore to develop analogs with varied efficacy, and potentially improved PK/PD, (ii) treatment of parasitic diseases with a single bolus dose of drug may provide more flexibility than dosing parameters established on the basis of cumulative side effects from repeated drug dosing.

2) The antiparasitic effect of ergotamine was accompanied by a marked apparent improvement in the pathological effects of infection on the host. The observed reduction in liver and spleen size and the improvement in liver morphology was found to correlate with an ergotamine-specific effect on hepatic stellate cells, and hence on liver fibrosis. A marked reduction in granuloma size and liver fibrosis is seen in congenitally athymic mice. However, in the case of S. mansoni infection, this leads to increased mortality and fewer eggs passed into the feces (Cheever et al., Am J Trop Med Hyg, 1993, 48:496-503). Byram and von Lichtenberg (Am J Trop Med Hyg, 1977, 26:944-56) suggested that the deficient granuloma formation led to failure to sequester toxic egg products, resulting in hepatocellular damage. Although smaller, the livers of ergotamine-treated, infected mice do not seem normal (Figure 5D) and have a darker color, even than control infected or PZQ-treated mice. Further, the picrosirius red-stained histological sections (Figure 5G) could be complemented with other staining methods to determine whether any hepatocyte necrosis is visible. It is important to show that the ergotamine effect on the host response is really beneficial.

We agree completely with these comments highlighting the multifaceted role of immune response to schistosome infection. As the reviewers point out, the granuloma may serve beneficial functions sequestering toxic egg components in the liver and facilitating egg transport across the intestinal mucosa for fecal excretion. If ergotamine treatment were to have deleterious effects, mimicking the scenario with T-cell deficient mice, we would expect to observe hepatocyte microvesicular steatosis and eggs accruing in the intestine (Doenhoff et al., 1986).

In the revised manuscript we have therefore expanded our histological analyses of control and drug-treated livers as suggested (Figure 6G, new text in the third paragraph of the subsection “In vivo efficacy of ergotamine”, new Figure 6—figure supplement 2 and 4). We agree that the gross appearance of ergotamine-treated livers is not identical to uninfected livers, but the expanded histology, taken together with the RNA-seq data, indicate that they do much more closely resemble the uninfected state than PZQ-treated samples. These data show that ergotamine treatment results in decreased liver collagen staining, decreased elastin staining, decreased lipid loss and no elevation in caspase activation relative to both infected and PZQ-treated samples (Figure 6G, Figure 6—figure supplement 2). As the reviewers note, impairing granuloma formation could have deleterious effects. However, histology does not reveal hepatocyte damage or obvious microvesicular steatosis in the livers of ergotamine treated mice that exceeds that already present in control infections (Figure 6—figure supplement 4). However, it is the case that all infected livers – from DMSO, ergotamine and praziquantel treated mice – exhibit pathology such as increased white blood cells and occasional slight “foaminess” within hepatocytes. Our proposed model of ergotamine inhibition of pro-fibrotic HSC activation would not necessarily replicate models of ablated host T-cell response, which show blockage of egg passage across the intestinal barrier. We did not find that ergotamine treatment resulted in an accrual of intestinal eggs (Figure 6C), but we also cannot state that this is due to functional immune-meditated transport since it likely largely reflects impaired egg production due to decreased worm burden (Figure 6B) and impaired egg development (Figure 6—figure supplement 1). These data, as well as the increased survival of ergotamine-treated animals, do not indicate that host pathology is worsened with ergotamine treatment. Figure 6 has been expanded and two additional figure supplements (Figure 6—figure supplements 2 and 4) have been added to reflect the additional histology performed. New text discussing these data has also been added (in the aforementioned subsection and subsection “Action on host”, second paragraph).

*3) Do these compounds (e.g., ergotamine) have activity against other schistosome stages (schistosomules, juveniles). Those experiments could be done* ex vivo *for a start, and subsequently* in vivo*. This issue is an important one, as juvenile worms (2-4 weeks p.i.) are refractory to PZQ. The issue should at least be discussed. Also do these compounds have activity in motility assays (or, even better,* in vivo*) against other species of schistosomes? Based on the data in Figure 3, one would expect they would, but it would be nice to know.*

We now provide both ex vivo and in vivo data to address this point in the revised manuscript. Figure 6—figure supplement 3 shows that ergotamine and 5-HT stimulate movement of juvenile worms ex vivo (Figure 6—figure supplement 3A and B). However in vivo, ergotamine – like PZQ – was ineffective at clearing infections (Figure 6—figure supplement 3C). Artemether was used as a positive control. These data suggest refractoriness to ergotamine at the juvenile stage of infection is not due to a lack of effect on the worms themselves. New text discussing these data has also been added (subsection “In vivo efficacy of ergotamine”, fourth paragraph).

4) Why is mouse survival lower in ergotamine-treated than in PZQ-treated infected mice despite the ameliorating effects of ergotamine on pathology.

Our in vivo work (Figure 6A) represents analysis of a single window of treatment (day 42-49) on survival over a subsequent 8 week period (to day 105). We expect that the remaining worm burden in mice sacrificed after 7 weeks reinitiates pathology over the subsequent monitoring period to cause the small extent of observed mortality.

[Editors' note: further revisions were requested prior to acceptance, as described below.]

The manuscript has been improved but there are some remaining issues that need to be addressed by editing the text before acceptance, as outlined below:In their revised version, Chan et al. addressed a number of points raised by the reviewers.With respect to my former comment that: "contraindications comprise among others hepatic disease, pregnancy, and the age of patients – such as preschoolers. Since schistosomiasis mainly affects children, including preschoolers, and since it is also a hepatic disease, how safe or how problematic could this drug be for a patient?" the authors responded that they "acknowledge the side effect profile of ergotamine and agree in the absence of preclinical data that speculation upon the human repurposing potential of ergotamine as a treatment is premature. […] Additionally, we note: (i) our SAR studies (Figure 3) underscore the potential for manipulation of the core ergot alkaloid pharmacophore to develop analogs with varied efficacy, and potentially improved PK/PD, (ii) treatment of parasitic diseases with a single bolus dose of drug may provide more flexibility than dosing parameters established on the basis of cumulative side effects from repeated drug dosing."In their marked version, however, it is not possible to follow these deletions since appropriate positions have not been marked as deleted within the text. Please indicate the "removed previous text speculating" clearly. Furthermore, I miss comments within the manuscript with respect to "the potential for manipulation of the core ergot alkaloid pharmacophore to develop analogs with varied efficacy, and potentially improved PK/PD". The chapter "Structural determinates of activity at schistosome 5-HTRs" ends with " ergotamine (compound '1) as the most potent, full agonist and 2-Bromo-LSD (compound '20') as the most potent antagonist for further evaluation." Unfortunately, also the Discussion fails to discuss this point critically. Instead, the Discussion ends with the conclusion that "[...] ergotamine is fortuitously suited to engage pathways in both host and parasite that are simultaneously deleterious to the blood fluke but beneficial to the infected host in terms of worm clearance and mitigation of damage caused by eggs" This is suggestive for its repurposing potential, even without mentioning this word.

Side effects of ergotamine:We apologize that the resubmitted version did not show deleted text in the comments pane, only the changed text, from the prior version. Sections deleted based on comments in the prior review report are now shown. We have added a new section in the Discussion (new text, subsection “Action on host”, third paragraph) which highlights the challenges of ergotamine as a lead compound. This new text makes clear the challenges with ergotamine as a repurposing candidate itself, which is not the intended implication of this study: ergotamine simply illustrates the potential benefit of coalescing beneficial host actions (on stellate cells) with deleterious antiparasitic activity in eliminating worms and mitigating their pathological effects. Ergotamine certainly has many additional side effects in healthy patients, which are fully acknowledged in the reviewers’ report and manuscript.

In the Abstract the authors mention that "Conventional approaches for antiparasitic drug discovery center upon discovering selective agents that adversely impact parasites with minimal host side effects". Especially the latter seems not applicable against the background of all known contradictions of ergotamine, its low bioavailability, and its limitations for children, which represent the major group of schistosomiasis patients.Therefore, the authors should critically discuss this point in their Discussion chapter and should mention that "analogs with varied efficacy, and potentially improved PK/PD" based on the ergotamine core are needed to develop a lead that may have the potential to become an effective and safe drug.

The final sentence of the Abstract has now been changed to underscore that ergotamine exemplifies a novel *strategy* for anthelmintic drug design by representing an agent that combines beneficial effects on host with deleterious actions on the parasite. The adverse side effects of ergotamine are explicitly addressed in new Discussion (subsection “Action on host”) that highlight predicted problems (the negative side effects observed with the beneficial host effects, low bioavailability, pediatric dosage).